# Polycomb repressive complexes 1 and 2 independently and dynamically regulate euchromatin during cerebellar neurodevelopment

Aditya Parmar[1], Anjali Srinivasan[1], Lena Krockenberger[1], Abijith Augustine[1], Owin Gong[1], Addison C. Bullard[1], Riya Kalra[1], Leya Ledvin[1], Dylan Pilz[1], Jonathan Tawil[1], Challana E. Tea[1], Kelly C. Wang[1], Olivia Urso[1], Larissa M. Kaube[1], Ying Sun[2], Roman Sasik[3], Kyle J. Gaulton[2], Kathleen M. Fisch[3,4], Cole J. Ferguson[1]*

1 Department of Pathology, University of California San Diego, La Jolla, California, United States of America, 2 Department of Pediatrics, University of California San Diego, La Jolla, California, United States of America, 3 Center for Computational Biology and Bioinformatics, University of California San Diego, La Jolla, California, United States of America, 4 Department of Obstetrics, Gynecology & Reproductive Sciences, University of California San Diego, La Jolla, California, United States of America

* cferguson@health.ucsd.edu

## Abstract

Polycomb Repressive Complexes (PRCs) are known for chemically modifying histones to compact chromatin structure and repress transcription. Broadly speaking, PRC1 monoubiquitinates histone 2A at lysine 119 (H2AK119ub), and PRC2 methylates histone H3 lysine 27 (H3K27me3, H3K27me2 and H3K27me1), but the scope and functions of these activities are complicated by a multiplicity of factors involving distinct cellular contexts and compositions of both complexes. Because epigenetic dysregulation is associated with neurodevelopmental disorders, but little is known about normal PRC activities in neurons, we used CUT&RUN to map PRC-dependent histone modifications in the mouse cerebellum at two postnatal timepoints (day 12 and 3 months). We find that H2AK119ub appears within both heterochromatin and euchromatin as the cerebellum matures, becoming enriched within active enhancers and promoters while being depleted from heterochromatin. Unexpectedly, the PRC1 product H2AK119ub appeared frequently without the accompaniment of the PRC2 product H3K27me3; leading to a much more dynamic chromatin state than when these two marks colocalized. Deposition of H2AK119ub at loci with the chromatin signature of active cis-regulatory elements tended to also gain the euchromatin-associated modifications H3K4me3 and H3K27ac during neurodevelopment. Importantly, deposition of H2AK119ub within both bivalent and H3K4me3-only promoters reduced transcription of downstream genes. The pattern of H2AK119ub deposition was specific to the cerebellum compared to liver and kidney. We then show that the PRC2 product H3K27me1 formed euchromatic zones that alternated with heterochromatic zones dominated by H3K27me3. Between the early and late timepoints H3K27me1 became enriched within a subset of expressed gene bodies and depleted

**Data availability statement:** Raw and analyzed data is available under the NCBI BioProject PRJNA1150596. All original code is available at: https://github.com/FergusonLab/CUT-RUN-pipeline.

**Funding:** The author(s) received no specific funding for this work.

**Competing interests:** The authors have declared that no competing interests exist.

from most other genes while remaining uncorrelated with the abundance of the corresponding mRNAs. Our data lead us to propose that deposition of H2AK119ub and H3K27me1 during cerebellar development likely fine-tunes the activity of cis-regulatory elements and transcription, respectively, and that PRC1 and PRC2 activities become uncoupled in the mature brain.

## Author summary

Brain development entails a very different set of cellular activities than healthy maturity: in early life neurons are born, differentiate, migrate, and integrate into neural circuits, while in maturity the neurons must maintain a fairly stable morphology while still being able to modulate synaptic connections and circuit performance in response to environmental cues and experience. These different functions in different contexts are made possible in part by activating and repressing distinct sets of genes through chromatin and epigenetic modifications. Because gaps persist in our understanding of how neuronal chromatin marks change between the postnatal period and adulthood, we mapped chromatin modifications produced by the Polycomb Repressive Complexes 1 and 2 (PRC1 and PRC2) during the maturation of the mouse brain. We uncover a surprising connection between PRC-dependent modifications and actively expressed genes. This association strengthens as the brain matures, while PRC-dependent modifications become depleted from repressed regions. We propose that PRC-dependent repressive modifications that appear in an activating context are used to fine-tune the degree of gene activation. Our findings carry fundamental implications for neurodevelopment and neurologic disease and suggest that the activities of PRC1 and PRC2 become uncoupled at many loci in the mature brain.

## Introduction

Cell fate decisions are made and maintained through the activation and repression of different genetic lineage programs [1–4]. Epigenetic regulation enables this task by recruiting factors to make various gene loci more or less accessible to the transcriptional machinery through open (euchromatin) or compacted (heterochromatin) structures [5]. In neurons, for example, different distributions of epigenetic marks are associated with cell cycle exit, morphogenesis, synaptogenesis, and maturity [6–11]. Yet these same cells also need to modify chromatin in a locus-specific fashion to enable stimulus-dependent plasticity [12,13]. It is therefore not surprising that epigenetic dysregulation due to environmental or genetic factors is associated with a variety of neurodevelopmental conditions, from autism spectrum disorders to epilepsy, schizophrenia, and inherited intellectual disability syndromes [14].

Epigenetic changes can affect the basic structural unit of chromatin, the nucleosome octamer, which consists of paired tetramers of four histone (H)

proteins—H2A, H2B, H3 and H4—enwrapped by DNA. Post-translational modifications (PTMs) of these histone proteins constitute the 'histone code' that helps determine a cell's identity and state [15]. Histone PTMs range from small chemical modifications such as methylation and acetylation [16] to ubiquitination, which is several orders of magnitude larger. Ubiquitin's size provides a larger binding platform for the recruitment of other factors which, in combination with ubiquitin itself, can then alter chromatin conformation in a multitude of ways. The most abundant and widely distributed form of monoubiquitinated histone is H2AK119ub (ubiquitination of histone 2A at lysine 119), which is deposited by the Polycomb Repressive Complex 1 (PRC1) [17,18]. Although there exist both canonical and noncanonical forms of PRC1 that contain different sets of accessory subunits imparting distinct functionalities [19–23], all PRC1 complexes in mammals include a heterodimer of ubiquitin ligase RING1 proteins that deposits H2AK119ub (hereafter H2Aub) [24] (Fig 1A). H2Aub leads the Polycomb Repressive Complex 2 (PRC2), which also has diverse compositional forms, to trimethylate histone 3 at lysine 27, generating the hallmark of cell type-specific gene repression in facultative heterochromatin (H3K27me3) [29–35].

Perturbances in PRC1 and PRC2 function are well-documented in cancers, but mutations in their interactors (e.g., *ASXL3*) or subunits (e.g., *EZH1*) have also been shown to disrupt neurogenesis, causing neurodevelopmental disorders associated with intellectual disability often accompanied by either overgrowth (e.g., Weaver syndrome, Cohen-Gibson syndrome) or small stature (e.g., Kabuki syndrome) [34–41]. Deficient function of either PRC1 or PRC2 has been linked to the pathogenesis of neurodegeneration [42,43]. The tumor suppressor *BAP1* (BRCA1-associated protein 1) [44], a histone deubiquitinase that refines the distribution of H2Aub [45–47] (Fig 1A), is mutated in an inherited syndrome involving speech, motor, and growth delays with neurobehavioral abnormalities [48]. Although studies of such conditions have provided a great deal of insight into epigenetic regulation during disease, the specific neuronal roles of the PRCs and the enzymatic products they uniquely control have not been comprehensively mapped genome-wide during neurodevelopment.

Because H2Aub is extensively reorganized during embryonic development [49,50], and PRC1 binding sites change dramatically when embryonic stems cells differentiate into neural progenitors [51], we wanted to understand how H2Aub might be redistributed over the course of postnatal neurodevelopment. Focusing on the cerebellum to take advantage of its cellular and neuronal homogeneity, we examined PRC-dependent histone modifications at two timepoints (P12 and 3 months). We find that in cerebellar neurons, H2Aub appears not only within sites of polycomb repression in facultative heterochromatin (i.e., chromatin that is compacted in specific cell types during development) but also, rather unexpectedly, within euchromatic loci. In fact, two patterns emerge: in one, H2Aub colocalizes with PRC2-mediated H3K27me3 (as expected), while another set of H2Aub marks colocalize with the activating marks H3K27ac and H3K4me3. This latter association becomes more prominent in the mature cerebellum. Collectively, our findings establish that a significant fraction of neuronal PRC1 activity occurs outside of the traditional context of heterochromatin, that PRC1-dependent epigenetic modifications in euchromatin are highly dynamic over the course of neurodevelopment, and that PRC1 may act independently of PRC2 in the mature brain.

## Results

### H2AK119ub is highly dynamic over the course of neurodevelopment and can act independently of H3K27me3

To capture changes that might occur in the genomic distribution of H2Aub in the mammalian brain over neurodevelopment, we examined the cerebellum at two timepoints. We chose the cerebellum because granule neurons constitute >85% of its cell population and >99% of its neurons, imparting a degree of homogeneity at the chromatin level [52] that facilitates ascertainment of chromatin states from bulk input. We chose P12 as the 'early' timepoint to maximize the representation of newly born neurons—granule neurons are produced in huge numbers from postnatal days 4–16 (P4 to P16) in mice [52,53]—and age 3 months as the 'late' timepoint, when the cerebellum reaches maturity [54].

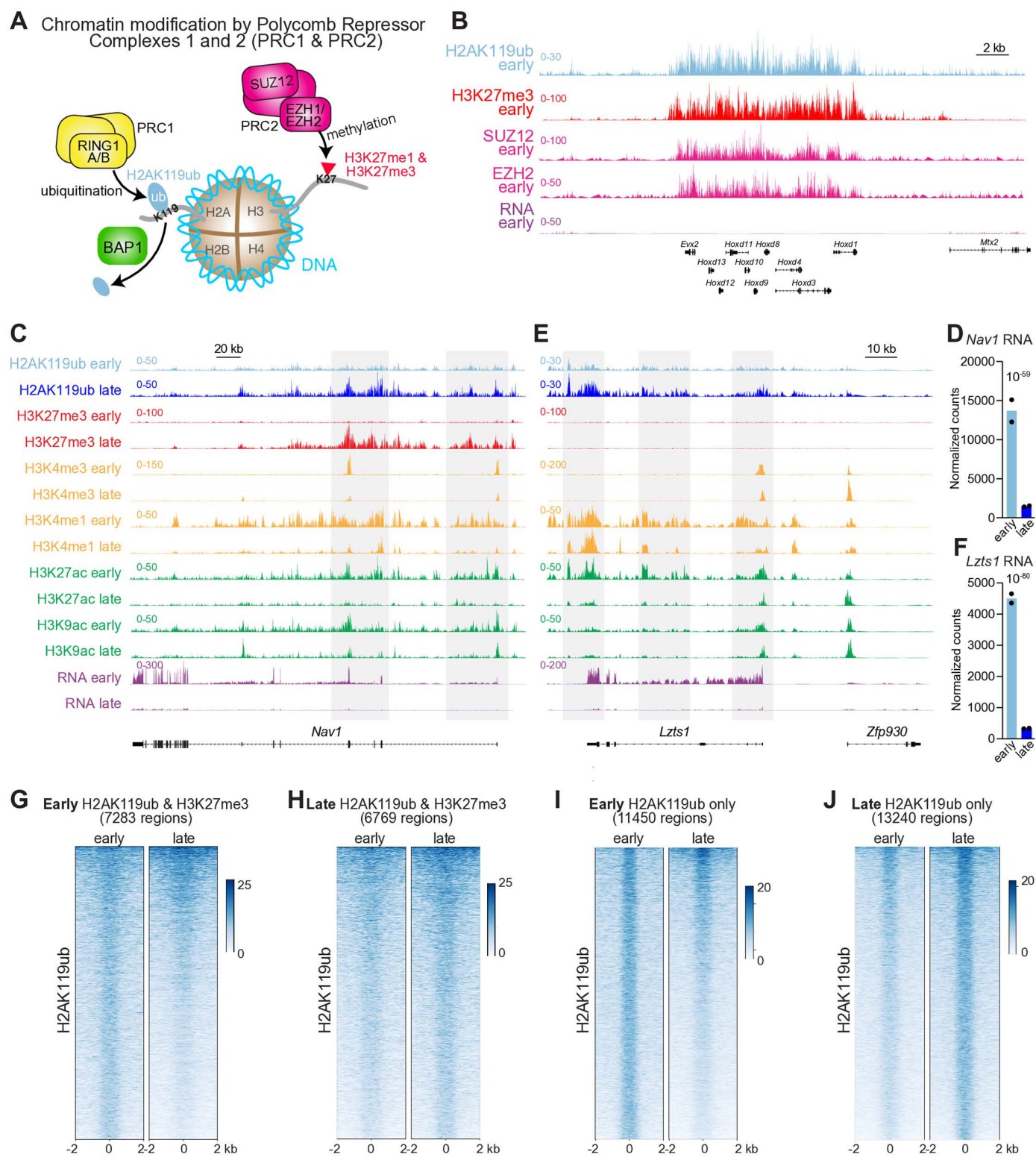

**Fig 1. The PRC1 product H2AK119ub can drive neurodevelopmental gene repression in the absence of PRC2-dependent H3K27me3.**
(A) Schematic of histone ubiquitination by the ubiquitin ligases RING1A and RING1B of the Polycomb Repressor Complex 1 (PRC1) to generate H2AK119ub, which is deubiquitinated by BAP1. Lys27 of histone 3 is methylated by the EZH1 and EZH2 histone methyltransferase subunits of the

Polycomb Repressor Complex 2 (PRC2). (B) CUT&RUN detection of H2AK119ub, H3K27me3, and the PRC2 components EZH2 and SUZ12 at the *HoxD* locus in nuclei isolated from the early (age P12) mouse cerebellum. RNAseq data is from ribodepleted RNA extracted from nuclei isolated from cerebellum at the early timepoint. (C) CUT&RUN tracks showing the indicated histone modifications alongside RNAseq tracks at the *Nav1* locus. Nuclei were isolated from early (P12) and late (3 month) mouse cerebellum. Shaded regions highlight neurodevelopmental deposition of H2AK119ub accompanied by loss of activating marks and gain of H3K27me3. (D) Normalized counts for the *Nav1* mRNA in RNAseq analysis of nuclei isolated from the early and late mouse cerebellum. Only reads mapping to exons were included. Normalization and adjusted p-value computed by DESeq2 [25] (n=2). (E) CUT&RUN tracks showing the indicated histone modifications alongside RNAseq tracks at the *Insm1* locus. Nuclei were isolated from early and late cerebellum. (F) DESeq2 normalized counts and adjusted p-value for the *Insm1* mRNA as detected by RNAseq analysis of nuclei isolated from the early and late mouse cerebellum (n=2). (G) Genome-wide aggregate heatmaps generated using deepTools [26] depicting normalized H2AK119ub CUT&RUN from the early and late mouse cerebellum. Plots are centered around loci that contain peak regions for both H2AK119ub and H3K27me3, as identified using SEACR [27] and intersected using BEDTools [28], in the early cerebellum. (H) Heatmaps of H2AK119ub CUT&RUN data centered around loci that contain peak regions for both H2AK119ub and H3K27me3, as identified in the late cerebellum. (I) Heatmaps of H2AK119ub CUT&RUN data centered around H2AK119ub peak regions identified in the early cerebellum that did not overlap with H3K27me3 peak regions. (J) Heatmaps of H2AK119ub CUT&RUN data centered around H2AK119ub peak regions identified in the late cerebellum that did not overlap with H3K27me3 peak regions.

We used CUT&RUN (Cleavage Under Targets and Release Using Nuclease), an antibody-based method known for its high resolution and low background [55,56], to detect H2Aub in nuclei isolated from the early and late cerebellum. For comparison, we also mapped H3K4me3 and H3K4me1 (enriched in euchromatin at promoters of actively transcribed genes and at active and poised enhancers, respectively), acetylated histones H3K27ac and H3K9ac (euchromatin; active enhancers and promoters), and their methylated counterparts H3K27me3 (facultative heterochromatin) and H3K9me3 (constitutive heterochromatin). We verified that antibodies to several of these modifications detected the appropriate modified nucleosomes from a panel spiked into experiments using cerebellar nuclei as input (S1A Fig). We modified the Cut&RunTools [57,58] pipeline to trim, align and call peaks using the SEACR (Sparse Enrichment Analysis for CUT&RUN) [27] and MACS2 (Model-Based Analysis of ChIP-seq 2) [59] algorithms (S1B Fig). To account for slight differences in sequencing depth, we developed a percentile-based normalization method which was conceptually similar to previous approaches for RNAseq [60] and ChIPseq [61]. After excluding a blacklist of 272 putative alignment artifacts, the algorithm identified local maxima in .bigwig files and recorded their heights. Data from one sample were then scaled by multiplying the height of every 50 bp bin by the ratio of the 99th percentile local maxima between the samples (S1C Fig). The data analyzed in this study for H2Aub and other modifications were sequenced to closely matched depths and typically required scaling by <10% to achieve high agreement between replicates (S1D and S1E Fig). To validate this method under more extreme conditions, we compared replicates that were sequenced at drastically different depths and still found a high degree of agreement (S1F and S1G Fig). Tracks depicting normalized H2Aub CUT&RUN data in four biological replicates of mouse cerebellum at the early and late timepoints (together constituting the eight data sets we went on to analyze genome-wide) demonstrated excellent agreement, across replicates and time points (S2A-S2C Fig).

As noted above, PRC1-dependent H2Aub has been studied primarily in the context of facultative heterochromatin, where it contributes to gene repression by recruiting PRC2 to generate H3K27me3, which inhibits RNA polymerase II [20,33,49,62,63]. In the early (P12) cerebellum, H2Aub overlapped with H3K27me3 within repressed loci such as the *HoxD* and *HoxA* gene clusters, which participate in early body planning but are subsequently silenced (Figs 1B and S3A). RNAseq of ribodepleted RNA extracted from nuclei isolated from the early cerebellum confirmed that these genes were not transcribed (Fig 1B). By including a brief cross-linking step after nuclear isolation, we were able to use CUT&RUN to detect the PRC2 subunits SUZ12 and EZH2 at the subset of heterochromatic loci with which PRC2 stably associates [64], such as *Hox* loci and the transcription factors *Prdm13, Foxl2* and *Osr2*, which are silenced in the developing cerebellum (Figs 1B and S3A-D). The distribution of PRC2 components at these repressed loci was similar to that of H2Aub.

We next compared the distribution of H2Aub to that of other histone modifications in the early and late (3 month-old) cerebellum at specific loci. In the early cerebellum, the gene *Nav1* (Neuron Navigator 1), which controls axon guidance, bore little H2Aub or H3K27me3 and instead harbored the euchromatin-associated modifications H3K4me3, H3K4me1, H3K27ac and H3K9ac (Fig 1C). By the late timepoint, however, the pattern was reversed: H2Aub became enriched while

each of the activating modifications was comparably depleted, such that the H2Aub distribution in the mature cerebellum bore a striking resemblance to CUT&RUN tracks of H3K27ac and H3K9ac in the early cerebellum. Collectively, these changes resulted in reduction of *Nav1* mRNA (Fig 1C and 1D). Additional early-expressed loci underwent silencing through the coordinated activity of PRC1 and PRC2 and the loss of activating modifications (S4A-S4D Fig). At the late timepoint H2Aub was accompanied by H3K27me3, but the latter was not strictly necessary for repression of early-expressed loci: genes such as the neurodevelopmental transcriptional regulator *Lzts1* lost their activating modifications and gained H2Aub without any concurrent increase in H3K27me3 (Fig 1E and 1F). This unexpected observation suggests that PRC1 and PRC2 can operate independently of one another in the developing brain.

This apparent uncoupling of PRC1 and PRC2 activities led us to examine H2Aub deposition in the two contexts, i.e., when it overlaps with H3K27me3 and when it does not. We used SEACR to call peak regions in H2Aub and H3K27me3 CUT&RUN data from the early and late cerebellum before using BEDTools [28] to identify regions in which these two histone modifications overlapped. At both timepoints, loci harboring H2Aub alone greatly outnumbered loci that possessed both H2Aub and H3K27me3. Using deepTools [26], we generated genome-wide aggregate heatmaps depicting normalized H2Aub data centered around loci harboring both marks at P12 (Fig 1G) or at 3 months (Fig 1H). Loci doubly marked at the early timepoint tended to lose H2Aub by 3 months (Fig 1G), whereas loci that bore both marks in the mature cerebellum had gained H2Aub (Fig 1H). In contrast, loci in the early cerebellum that harbored only H2Aub showed a conspicuous decline in H2Aub by 3 months (Fig 1I), while loci in the mature cerebellum harboring H2Aub alone had undergone an obvious increase in this histone modification (Fig 1J). Thus, not only does PRC1 seem able to operate independently of PRC2, but PRC1-dependent H2Aub is more dynamic when not affiliated with H3K27me3.

## Cis-regulatory elements show the greatest changes in H2AK119ub over neurodevelopment

To begin to understand how H2Aub changes on a genome-wide scale, we analyzed its reorganization during cerebellar maturation within different regulatory contexts. We first identified loci in which H2Aub changed significantly between the two timepoints by subjecting four biological replicates of H2Aub CUT&RUN data from the early and late cerebellum to genome-wide statistical analysis using DiffBind [65,66]. Principal component analysis (PCA) separated datasets from each timepoint along the major axis (principal component 1), which accounted for 74% of the observed variability (Fig 2A). SEACR identified 39358 consensus peak regions across all datasets. Setting the false-discovery rate to <0.05, DiffBind identified 6246 (15.9%) peak regions in which H2Aub significantly increased, and 7967 (20.2%) where it significantly decreased, between the early and late timepoints. Volcano plots and minus-average plots, which depict the ratio of late to early H2Aub versus significance or H2Aub abundance (concentration), respectively, showed that a similar number of regions experienced significant H2Aub enrichment as depletion (Fig 2B and 2C). Heatmaps depicting normalized H2Aub data centered on these three classes of loci—unchanged, significantly up or significantly down—accorded with predictions from the DiffBind output and established the foundation for subsequent analyses of other histone modifications within regions exhibiting distinct patterns of developmental H2Aub deposition (Fig 2D).

Using the annotation tool HOMER (Hypergeometric Optimization of Motif EnRichment), [67] we found that most H2Aub peak regions occurred within introns, followed by intergenic regions and promoters (Fig 2E). Gains in H2Aub were more likely to occur at promoters, whereas losses were more likely to occur within intergenic regions. Introns and intergenic regions typically harbor non-promoter cis-regulatory elements such as active and repressed enhancers, loci typically marked by H3K4me1 [68]. Active enhancers are typically further modified by acetylation (e.g., H3K27ac and H3K9ac) [69,70], whereas repressed enhancers are further modified by H3K27me3. We observed localization of H2Aub to both repressed and active enhancers (Fig 2F and 2G). These results imply that H2Aub primarily localizes to regulatory elements rather than gene bodies.

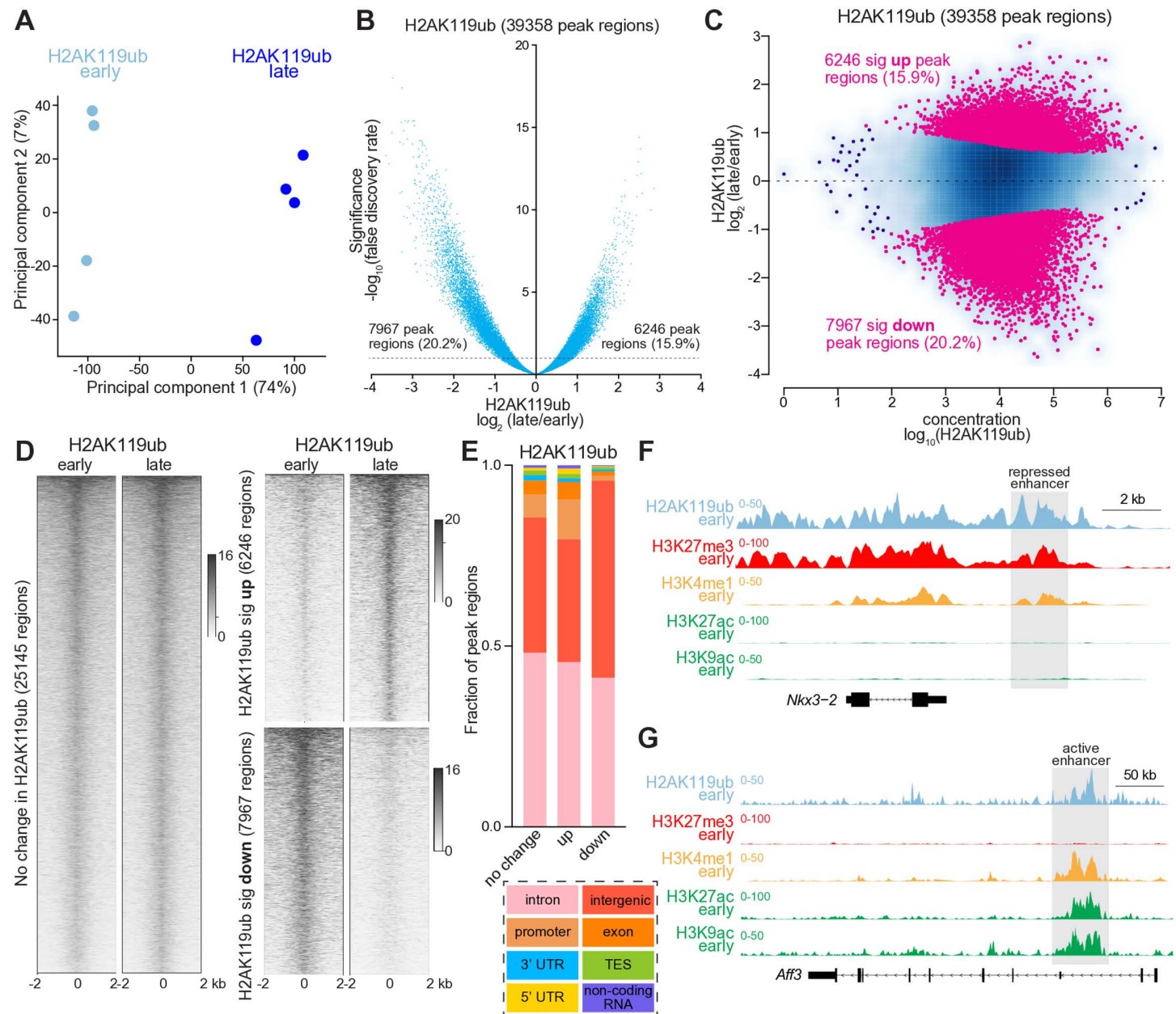

**Fig 2. H2AK119ub is bidirectionally dynamic within cis-regulatory elements during neurodevelopment. (A)** Principal component analysis (PCA) of H2AK119ub CUT&RUN data from early and late cerebellum (n = 4). **(B)** Volcano plot depicting H2AK119ub abundance, as detected in normalized CUT&RUN data from early and late cerebellum, within SEACR-defined consensus peak regions. The significance threshold was a false-discovery rate <0.05, as defined by DiffBind [65] (n = 4). **(C)** Minus-average (MA) plot comparing H2AK119ub CUT&RUN data from early and late cerebellum within H2AK119ub consensus peak regions, as quantitated using DiffBind (n = 4). **(D)** Heatmaps depicting normalized H2AK119ub CUT&RUN data in the early and late cerebellum centered around loci with unchanged, increased or decreased H2AK119ub, as defined by DiffBind. **(E)** Functional annotation of H2AK119ub peak regions using HOMER [67]. **(F)** CUT&RUN tracks showing a repressed enhancer from the early cerebellum. **(G)** CUT&RUN tracks showing an active enhancer from the early cerebellum.

## H2AK119ub is depleted from facultative heterochromatin as the cerebellum matures

After observing that H2Aub is associated with both heterochromatin and euchromatin in the brain and that its distribution changes considerably during cerebellar neurodevelopment, we wondered whether there was a predominant pattern within these distinct chromatin contexts. We knew that some heterochromatic loci that harbored H3K27me3 in the early cerebellum gained H2Aub, while others were depleted (Fig 3A). To analyze these changes within facultative heterochromatin genome-wide, we used SEACR to identify H3K27me3 peak regions in the early and late cerebellum and then calculated the abundance of H2Aub within these regions. To permit comparisons between regions, we scaled the abundance of H2Aub to the length of each peak region. We found that H2Aub abundance declined in facultative heterochromatin over the course of cerebellar maturation, with Kolmogorov-Smirnov D statistic (which compare differences between cumulative distributions) equal to 0.23 for early H3K27me3 peak regions and 0.21 for late H3K27me3 peak regions (Fig 3B). To visualize this effect, we used the MACS2 narrow algorithm to identify peak summits in H3K27me3 CUT&RUN data before generating heatmaps depicting normalized H2Aub data centered around these peaks. Whether oriented around H3K27me3 peaks from the early or late cerebellum, H2Aub concentration fell (Figs 3C and S5A).

To determine whether developmental changes in H2Aub were statistically significant, we compared the length-scaled abundance of H2Aub in four biological replicates using edgeR [71] and limma [72], tools that permit the input of normalized data. Volcano plots depicting the ratio of H2AK119ub (late/early) in the cerebellum showed a leftward skew within H3K27me3 peak regions in both the early cerebellum (Fig 3D) and late cerebellum (S5B Fig). Setting a threshold of 1.5x fold-change in H2Aub abundance within each peak region, and an adjusted p-value <0.05, we found that many more peak regions lost H2Aub than gained it.

To determine whether H2Aub and H3K27me3 were coregulated within heterochromatin during cerebellar maturation, we compared the ratio (late/early) of both modifications within H3K27me3 peak regions from the early or late cerebellum. These $\log_2$-$\log_2$ plots revealed no clear correlation between the directionality or scale of changes in these modifications (Figs 3E and S5C). Plotting H3K27me3 CUT&RUN data within loci that underwent significant increases or decreases in H2Aub did not reveal comparable changes in H3K27me3, which remained relatively static in all three classes of loci (Fig 3F). Within loci that harbored both H2Aub and H3K27me3, H3K27me3 was also more stable across cerebellar development than within loci harboring H2Aub alone (Fig 3G and 3H). Interestingly, at loci where H2Aub appeared in the absence of H3K27me3, H3K27me3 changed in the opposite direction as H2Aub: loci identified in the early cerebellum lost H2Aub and gained H3K27me3 (Figs 1I and 3H), whereas loci identified in the late cerebellum gained H2Aub and lost H3K27me3 (Figs 1J and 3H). These observations indicate that PRC1 and PRC2 operate independently of one another at most loci as the cerebellum matures.

## H2AK119ub becomes enriched within H3K4-methylated loci and CpG islands

We next used SEACR to identify H3K4me3 peak regions, typically enriched in promoters of expressed genes, in the early and late cerebellum before measuring the length-scaled abundance of H2Aub within these regions. The late cerebellum contained significantly more H2Aub (Fig 4A). Aggregate heatmaps show H2Aub becomes enriched over time in relation to H3K4me3 peak summits (Figs 4B and S6A) as well as within bivalent promoters, i.e., H3K4me3-labeled promoters that also harbor H3K27me3 and encode proteins involved in maturation of cerebellar granule neurons [73,74] (Fig 4C). To determine whether these changes were significant, we statistically analyzed biological replicates as described above. Volcano plots of H2Aub data demonstrated a clear rightward skew, whether compared against H3K4me3 peak regions in the early (Fig 4D) or late cerebellum (S6B Fig): in either case, only a few dozen regions experienced a significant reduction in H2Aub. These findings demonstrate that H3K4me3-positive chromatin becomes more ubiquitinated as the brain matures.

To understand how H3K4me3 changed in relation to H2Aub, we plotted H3K4me3 CUT&RUN data centered around previously defined loci that underwent no change, an increase, or a decrease in H2Aub over cerebellar development.

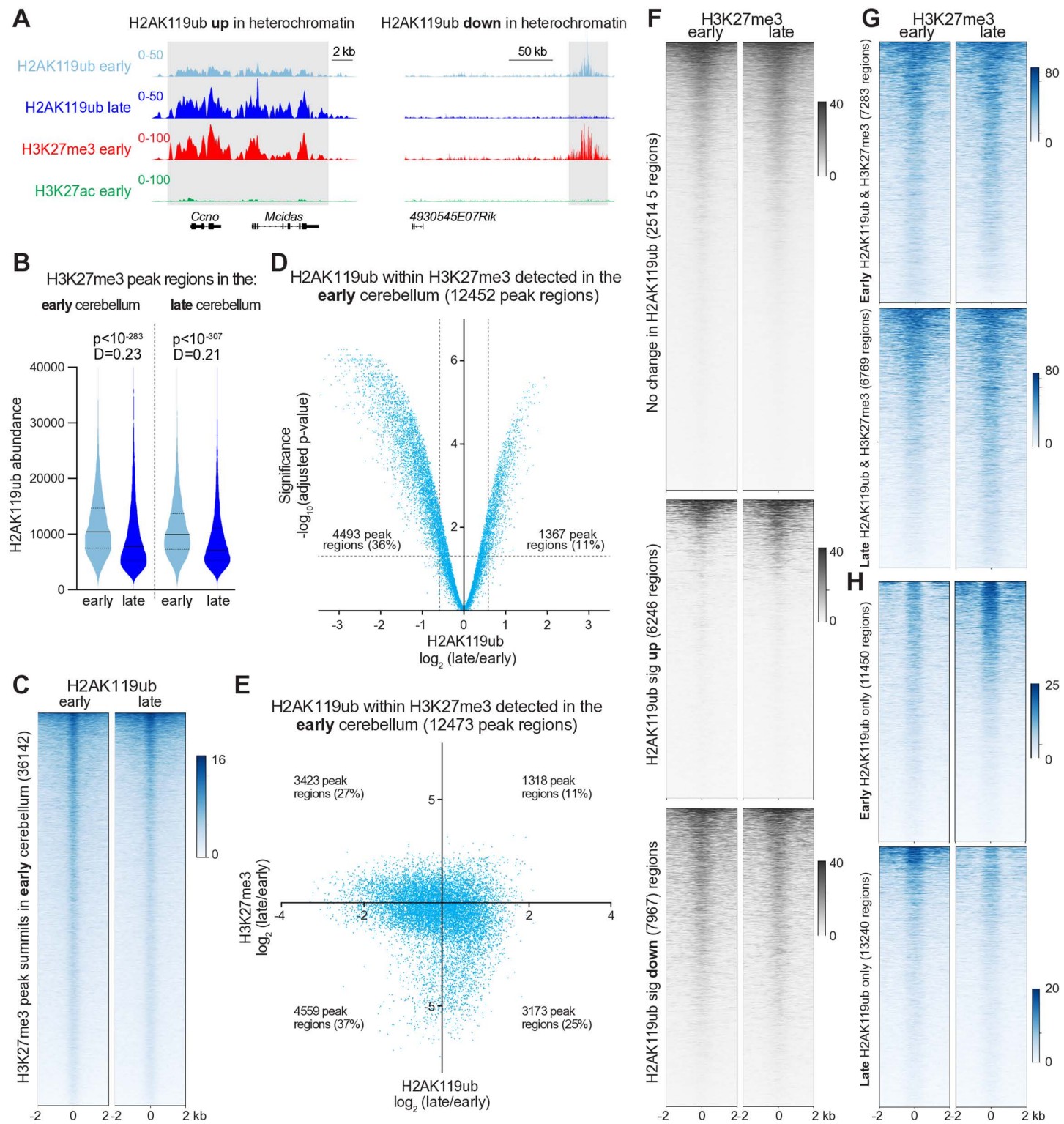

**Fig 3. H2AK119ub is depleted from facultative heterochromatin as the cerebellum matures. (A)** CUT&RUN detection of the indicated modifications at the indicated timepoints. **(B)** Violin plots showing length-scaled abundance of H2AK119ub within SEACR-defined H3K27me3 peak regions identified in the early and late cerebellum. P-values and distance (D) statistics by Kolmogorov-Smirnov (K-S) test (n = 4). **(C)** Heatmaps depicting H2AK119ub CUT&RUN data in the early and late cerebellum centered around H3K27me3 peak summits identified by MACS2 [59] narrow in the early cerebellum.

In general, the number of peak summits detected is greater than the number of peak regions, as each region tended to harbor multiple summits. **(D)** Volcano plot depicting H2AK119ub abundance, as detected in normalized CUT&RUN data from early and late cerebellum, within H3K27me peak regions identified by SEACR in the early cerebellum. The significance threshold was a p-value <0.05, as computed using edgeR [71] and limma [72] (n = 4). **(E)** Log$_2$-log$_2$ scatterplot comparing the ratio of H2AK119ub (late/early) to the ratio of H3K27me3 (late/early) within H3K27me3 peak regions identified in the early cerebellum (n = 4 for H2AK119ub, n = 2 for H3K27me3). **(F)** Heatmaps depicting H3K27me3 CUT&RUN data centered around previously defined loci with unchanged, increased or decreased H2AK119ub. **(G)** Heatmaps depicting H3K27me3 within loci harboring peak regions for both H2AK119ub and H3K27me3 at the early (*top*) and late (*bottom*) timepoints. **(H)** Heatmaps depicting H3K27me3 within loci harboring peak regions for H2AK119ub in the absence of H3K27me3 at the early (*top*) and late (*bottom*) timepoints.

H3K4me3 became more abundant within a subset of each the first two classes of loci (Fig 4E), but it was lost over development within loci where H2Aub declined. Log$_2$-log$_2$ plots depicting changes in H3K4me3 and H2Aub within H3K4me3 peak regions demonstrate that both modifications increased over time in most of these loci (62% of peak regions from early cerebellum, 77% of peak regions from late cerebellum) (S6C and S6D Fig). H3K4me3 also increased within loci harboring H2Aub in combination with H3K27me3 (Fig 4F). Strikingly, however, it was loci that harbored H2Aub alone, particularly those detected at the late timepoint, where H3K4me3 was both most abundant and most conspicuously enriched (Fig 4G). These observations indicate that both H2Aub and H3K4me3 undergo a coordinated program of enrichment within active promoters as the cerebellum matures.

To determine how promoter-associated H2Aub deposition affects transcription, we analyzed mRNA abundance over neurodevelopment within protein-coding genes with different types of promoters. When plotted in metagene heatmaps, genes whose promoters were labeled by H3K4me3 alone displayed no obvious change in mRNA abundance during cerebellar maturation (Fig 4H), nor did genes with bivalent promoters (Fig 4I). However, genes whose promoters (both H3K4me3-only and bivalent) experienced a significant increase in H2Aub during cerebellar maturation exhibited reduction in mRNA (Fig 4J and 4K), an effect which was not observed among promoters that experienced no change in H2Aub (Fig 4L and 4M). Deposition of H2Aub at promoters thus subdues transcription during neurodevelopment.

We next examined H2Aub within all 17017 CpG islands, which are promoter-associated DNA sequences that represent a major target of PRC1 activity during the establishment of polycomb domains [75]. Some CpG islands acquired H2Aub during cerebellar maturation (e.g., *Bcat1,* which encodes the catabolic enzyme branched-chain amino acid transaminase 1) (S7A Fig) while others lost it (e.g., *Isl1,* which encodes the developmental transcription factor Insulin gene enhancer protein 1) (S7B Fig). Overall, however, H2Aub increased over time (S7C Fig), and a greater fraction of H2Aub peak regions from the late cerebellum overlapped with CpG islands than did early peak regions (S7D Fig). Heatmaps centered around CpG islands also showed an increase in H2Aub over neurodevelopment (S7E Fig).

To determine how histone ubiquitination changes within non-promoter cis-regulatory elements, namely, enhancers marked by H3K4me1, which may be either active or repressed, we examined H3K4me1 peak regions detected in the early cerebellum. H2Aub changed much less in these regions than it did within H3K4me3 peak regions (S8A Fig). Genome-wide aggregate heatmaps centered around H3K4me1 peak summits did not reveal appreciable gains in H2Aub over time (S8B Fig). Within H3K4me1 peak regions identified in the late cerebellum, however, H2Aub did increase over time (S8A Fig). Genome-wide aggregate plots centered around late H3K4me1 peak summits showed a slight increase in H2Aub (S8C Fig). These data suggest that H2Aub plays a larger role in cis regulation in the mature cerebellum.

## Many active enhancers in the mature cerebellum gain both H2AK119ub and H3K27ac

After observing enrichment within active promoters, we sought to understand how neurons remodel H2Aub within euchromatin, which is primarily defined by the presence of histone acetylation. We concentrated on H3K27ac and H3K9ac because these marks are the best studied and have highly specific antibodies. While certain loci that harbored H3K27ac in the early cerebellum might gain or lose H2Aub (Fig 5A), H3K27ac peak regions overall gained H2Aub during cerebellar development (Fig 5B). Heatmaps centered on H3K27ac peak summits also showed increased H2Aub (Figs 5C and S9A).

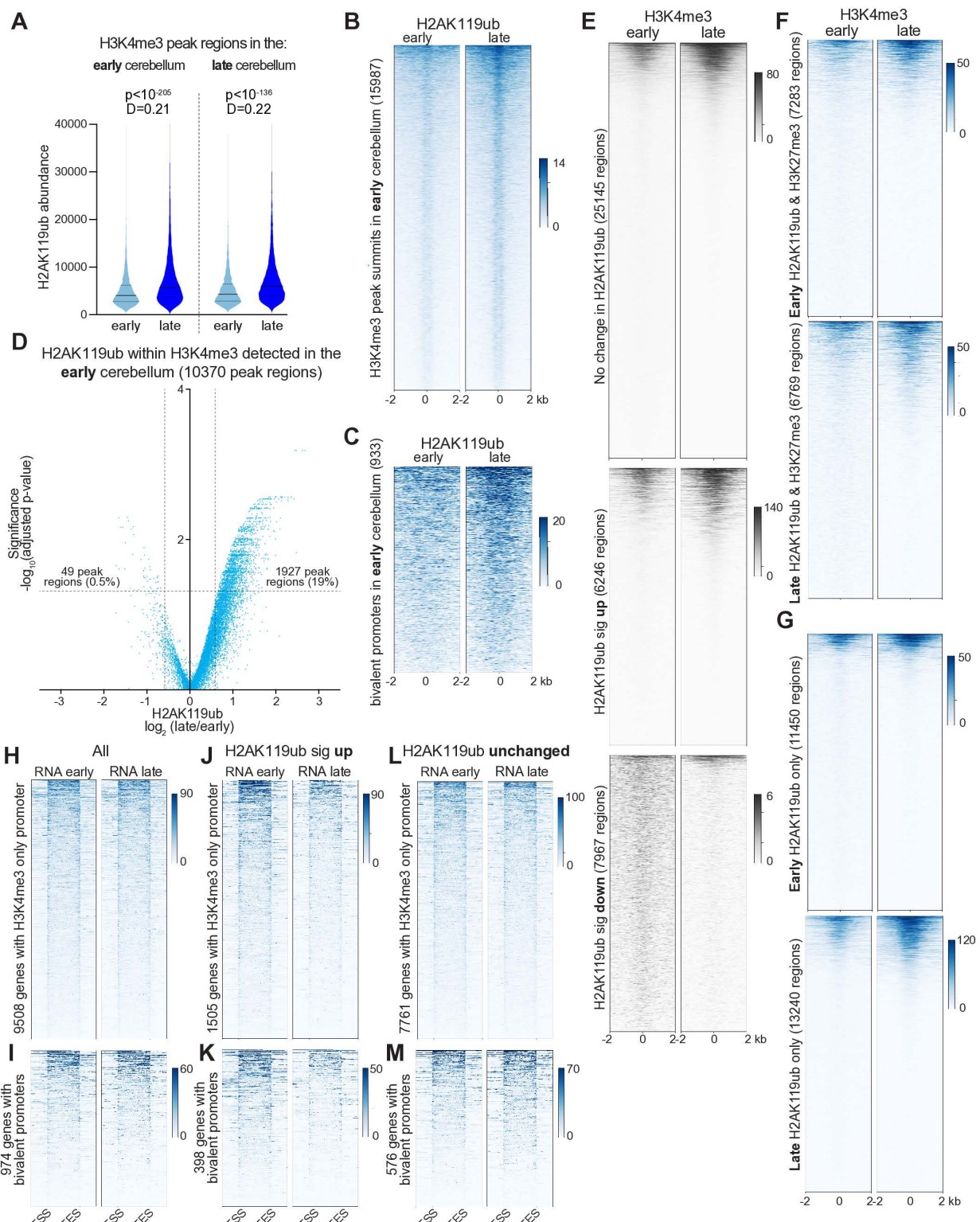

**Fig 4. Neurodevelopmental H2AK119ub enrichment within H3K4me3-positive promoters subdues transcription. (A)** Violin plots showing length-scaled abundance of H2AK119ub within H3K4me3 peak regions, P-values and D statistics by Kolmogorov-Smirnov D test (n = 4). **(B)** Heatmaps of normalized H2AK119ub CUT&RUN data centered around H3K4me3 peak summits identified by MACS2 narrow in the early cerebellum. **(C)** Heatmaps of CUT&RUN H2AK119ub signal at bivalent promoters identified in the early cerebellum. Bivalent promoters were defined by intersecting peak

regions identified for H3K4me3 and H3K27me3 in the early cerebellum. **(D)** Volcano plot depicting H2AK119ub abundance, as detected in normalized CUT&RUN data from early and late cerebellum, within H3K4me3 peak regions identified by SEACR in the early cerebellum. The significance threshold was a p-value <0.05, as computed using edgeR and limma (n = 4). **(E)** Heatmaps depicting H3K4me3 CUT&RUN data centered around loci with unchanged, increased or decreased H2AK119ub. **(F)** Heatmaps depicting H3K4me3 CUT&RUN data within loci harboring peak regions for both H2AK119ub and H3K27me3 at the early and late timepoints. **(G)** Heatmaps depicting H3K4me3 CUT&RUN data within loci harboring peak regions for H2AK119ub in the absence of H3K27me3 at the early and late timepoints. **(H)** Metagene heatmaps depicting RNAseq data from the early and late cerebellum within genes possessing H3K4me3-only promoters, as detected in the early cerebellum. **(I)** RNAseq data within genes possessing bivalent promoters harboring peak regions for both H3K4me3 and H3K27me3, as detected in the early cerebellum. **(J)** RNAseq data within genes possessing H3K4me3-only promoters that experienced a significant increase in H2AK119ub. **(K)** RNAseq data within genes possessing bivalent promoters that experienced a significant increase in H2AK119ub. **(L)** RNAseq data within genes possessing H3K4me3-only promoters that experienced no change in H2AK119ub. **(M)** RNAseq data within genes possessing bivalent promoters that experienced no change in H2AK119ub.

Aggregate plots centered on active enhancers (defined as the intersection of peak regions for H3K27ac and K4me1) showed H2Aub deposition over time (Figs 5D and S9B). To determine whether these changes were statistically significant, we analyzed biological replicates as above: volcano plots exhibited rightward skew, and many more regions gained H2Aub than lost it (Figs 5E and S9C). Some H3K27ac peak regions that exhibited the greatest increase in H2Aub resided within clusters of active enhancers, each of which became ubiquitinated to a much greater degree at the late timepoint and correlated with RNA suppression (S9D Fig).

To determine whether H2Aub and H3K27ac were coordinately regulated within euchromatin, we compared the ratio (late/early) of the length-scaled abundance of each modification within H3K27ac peak regions identified in the early cerebellum. $\text{Log}_2\text{-log}_2$ plots demonstrate that 61% of peak regions gained H2Aub while losing H3K27ac (Fig 5F), which demonstrates that ubiquitination is part of a broader repressive program that simultaneously depletes euchromatin-associated modifications from loci that are active in the early cerebellum. However, when we compared H2Aub to H3K27ac within H3K27ac peak regions detected in the late cerebellum, the data were shifted noticeably upward, with 43% of loci gaining both H2Aub and H3K27ac (Fig 5G). These findings suggest that H2Aub is likely to function differently at enhancers that are active during neuronal development versus those that are active in mature neurons. Aggregate plots depicting H3K27ac CUT&RUN data in relation to sites with no change, increased or decreased H2Aub showed a developmental increase in the first two classes, where H3K27ac was abundant (Fig 5H). For example, active enhancers near the *Potefam3a* locus or within the *Tet3* locus gained both H2Aub and H3K27ac without any change in H3K27me3 (Figs 5I and S9E). Similar to H3K4me3, H3K27ac was not very abundant and declined even further within sites that lost H2Aub during neurodevelopment (Fig 5H). Within sites where H2Aub and H3K27me3 overlapped, H3K27ac declined over time and was much less abundant than within sites harboring H2Aub without H3K27me3, where H3K27ac increased (Fig 5J and 5K). Thus, H2Aub appearing outside of the traditionally defined heterochromatic (H3K27me3-positive) context becomes increasingly associated with H3K27ac as the cerebellum matures.

H3K9ac peak regions, which often surround transcription start sites, saw increases of H2Aub (S10A Fig), and heatmaps centered on H3K9ac peak summits demonstrated developmental gains of H2Aub (S10B and S10C Fig). Testing the statistical significance of changes in H2Aub in biological replicates skewed volcano plots to the right, with many more regions experiencing increases in H2Aub than decreases (S10D and S10E Fig). Within loci where H2Aub was unchanged or increased over time, H3K9ac—like H3K27ac and H3K4me3—was not very abundant. At sites where H2Aub decreased, however, H3K9ac was quite abundant in the early cerebellum but declined dramatically by the late timepoint (S10F Fig). Interestingly, unlike H3K27ac and H3K4me3, H3K9ac did not change within loci that harbored H2Aub alone or in combination with H3K27me3 (S10G and S10H Fig).

## H2AK119ub is enriched in euchromatin and depleted from heterochromatin during neurodevelopment

To develop a more holistic understanding of how H2Aub remodeling relates to other key histone modifications during cerebellar maturation, we computed Pearson correlation coefficients between H2Aub and other histone PTMs as detected

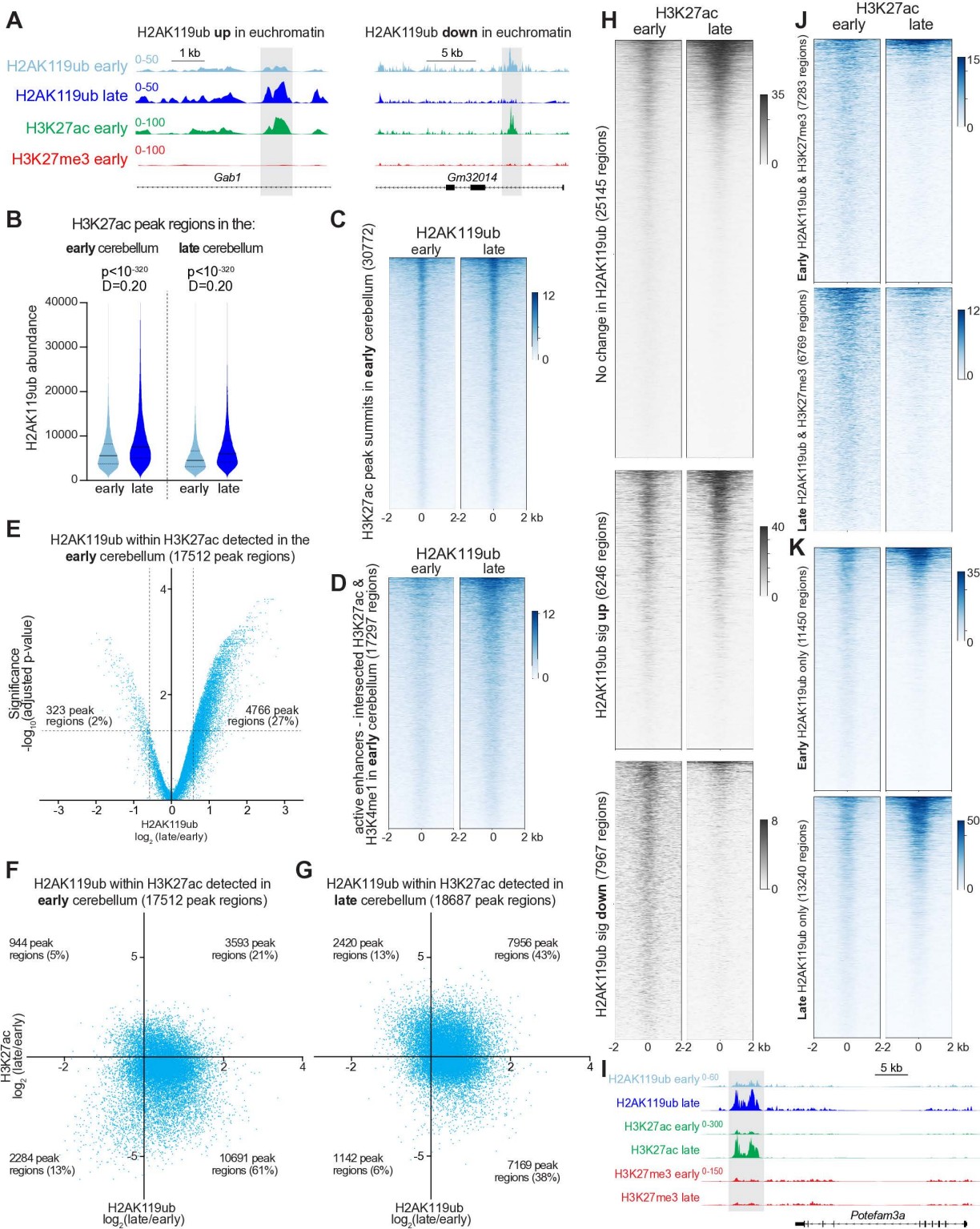

**Fig 5. Neurodevelopmental deposition of H2AK119ub within acetylated euchromatin. (A)** CUT&RUN detection of the indicated modifications at the indicated timepoints. **(B)** Violin plots showing length-scaled abundance of H2AK119ub within H3K27ac peak regions. P-values and D statistics by K-S D test (n = 4). **(C)** Heatmaps of normalized H2AK119ub CUT&RUN centered around H3K27ac peak summits identified by MACS2 narrow in the early cerebellum. **(D)** Heatmaps of H2AK119ub CUT&RUN data centered around active enhancers (intersection of SEACR-defined H3K27ac peak regions and

MACS2 broad-defined H3K4me1 peak regions) identified in the early cerebellum. **(E)** Volcano plot depicting H2AK119ub in the early and late cerebellum, as quantified from normalized CUT&RUN data, within H3K4me3 peak regions identified the early cerebellum. The significance threshold was a p-value <0.05, as computed using edgeR and limma (n = 4). **(F)** $Log_2$-$log_2$ scatterplot comparing changes in H2AK119ub and H3K27ac within H3K27ac peak regions detected in the early cerebellum. **(G)** $Log_2$-$log_2$ plot comparing changes in H2AK119ub and H3K27ac within H3K27ac peak regions detected in the late cerebellum. **(H)** Heatmaps depicting H3K27ac CUT&RUN data centered around loci with unchanged, increased, or decreased H2AK119ub. **(I)** CUT&RUN data for the indicated histone modifications at the indicated timepoints. **(J)** Heatmaps depicting H3K27ac CUT&RUN data within loci harboring peak regions for both H2AK119ub and H3K27me3 at the early and late timepoints. **(K)** Heatmaps depicting H3K27ac CUT&RUN data within loci harboring peak regions for H2AK119ub in the absence of H3K27me3 at the early and late timepoints.

by CUT&RUN in biological replicates of early and late mouse cerebellum (Fig 6A). As expected, at both timepoints the activating modifications H3K4me3, H3K4me1, H3K27ac and H3K9ac correlated with one another but not with H3K27me3. Compared to the other modifications, H2Aub underwent greater changes between the early and late timepoints, resulting in a more modest correlation (0.43 and 0.47 in two biological replicates) (Fig 6B). In agreement, the fraction of peak regions that overlapped between the early and late timepoints was lower for H2Aub than the other modifications (Fig 6C). Importantly, the correlation between H2Aub and all the activating modifications increased over time, regardless of whether we considered them at the early or the late timepoint (Fig 6D), while the correlation between H2Aub and the repressive mark H3K27me3 weakened over time. The fraction of peak regions that overlapped with those identified for H3K4me3, H3K4me1, H3K27ac and H3K9ac were all higher than the portion of H2Aub peak regions detected in the late cerebellum, a trend that was not observed when comparing H2Aub with H3K27me3 (S11 Fig). Interestingly, at both early and late stages of cerebellar development, H2Aub overlapped to the greatest degree with the enhancer-associated modification H3K4me1. These observations suggest that, to a large degree, active loci in the early cerebellum gain H2Aub over the course of cerebellar maturation.

Next, we employed Hidden Markov Modeling using ChromHMM [76], inputting CUT&RUN data from the early or late cerebellum to segment the genome into functional states and determine their association with H2Aub. In addition to the aforementioned histone PTMs, we also included H3K9me3 for its association with constitutive heterochromatin, and the PRC2 product H3K27me1 for its association with gene bodies of transcribed loci [77]. Separate models were generated using data from just the early cerebellum or just the late cerebellum. At both timepoints, 15-state HMMs identified major states of chromatin, correctly distinguishing active transcription (dominated by H3K27me1) and constitutive heterochromatin (dominated by H3K9me3) (Fig 6E and 6F). In the early cerebellum, H2Aub was most strongly associated with weak active enhancers, followed by repressed enhancers and facultative heterochromatin (Fig 6E). In the late cerebellum, however, H2Aub was most associated with strong active enhancers (i.e., loci which harbored H3K4me1, H3K27ac and H3K9ac but not H3K4me3 or H3K27me3) (Fig 6F). These genome-wide analyses demonstrate that H2Aub operates within diverse chromatin contexts but shifts its distribution away from heterochromatin in the early cerebellum to become more associated with euchromatin in the late cerebellum.

### H2AK119ub contributes to tissue-specific chromatin states at both active and repressed loci

After characterizing the contribution of PRC1-dependent H2Aub to changes in chromatin state that occur during maturation of a single tissue type (the cerebellum), we set out to examine H2Aub in relation to the second major purpose of histone modifications: establishing tissue-specific patterns of epigenetic gene regulation. Histone modifications at cis-regulatory elements are a central mechanism by which cells bring about distinct tissue-specific chromatin states [78], but this comparison has never been made for H2Aub. We therefore compared the patterns of H2Aub deposition between the cerebellum, liver and kidney at the late timepoint (age 3 months). Certain loci showed shared patterns of H2Aub, H3K27ac and H3K27me3 deposition across tissues, whereas other loci exhibited cerebellum-specific patterns of enrichment or depletion for each modification (Fig 7A-7C). In terms of its overall distribution, neuronal H2Aub peak regions resided more

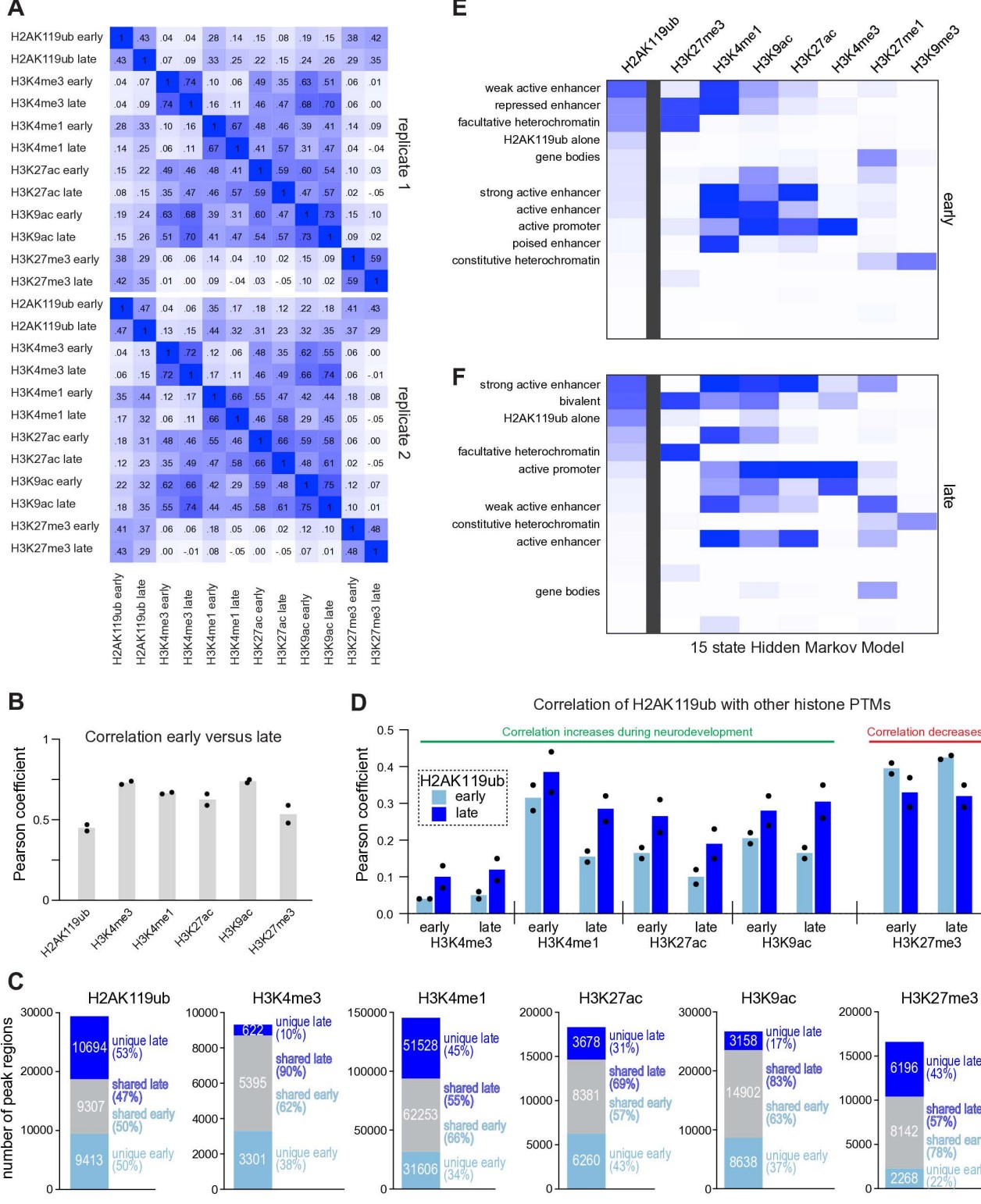

**Fig 6. H2AK119ub is enriched in euchromatin and depleted from heterochromatin during neurodevelopment. (A)** Correlation matrices showing the Pearson correlation coefficient for CUT&RUN data in normalized biological replicates. **(B)** Bar graph comparing the Pearson coefficient for histone modifications in CUT&RUN data from the early versus late cerebellum (n = 2). **(C)** Stacked bar chart comparing the fraction of overlapping peak regions

for different histone modifications in the early versus late cerebellum. **(D)** Bar graphs comparing the Pearson correlation between H2AK119ub and other histone modifications in the early and late cerebellum. (E) 15-state Chromatin Hidden Markov modeling (ChromHMM [76]) of CUT&RUN data from the early cerebellum, ordered according to the representation of H2AK119ub (n = 2 replicates for each histone modifications). **(F)** ChromHMM analysis of CUT&RUN data from the late cerebellum (n = 2).

frequently within intergenic regions and introns (locations typically associated with enhancers), whereas H2Aub in the liver and kidney more frequently appeared within promoters and exons (Fig 7D).

Pearson correlation coefficients reveal how H2Aub corresponds to H3K4me3, H3K4me1, H3K27ac, H3K9ac and H3K27me3 in the adult liver and kidney (Fig 7E). H2Aub correlated with both activating and repressive histone modifications in the liver and kidney, indicating that H2Aub is associated with euchromatin in non-neural organs as well as in the brain. For the histone modifications H2Aub, H3K4me1, H3K27ac and H3K27me3, the liver and kidney correlated more with each another than either did with the cerebellum (Fig 7F). As an additional method of comparing the genomic pattern of histone PTMs between different tissues, we called peak regions and used BEDTools [28] to compute the fraction of overlapping peak regions for H2Aub, H3K27ac and H3K27me3 between the cerebellum and non-neural organs (S12A-S12I Fig). The resulting Venn diagrams demonstrated that most peaks for all three histone PTMs were tissue-specific, although a sizable minority were shared across tissues (S12J Fig). Gene ontology (GO) analysis of the genes residing in closest proximity to cerebellum-specific H2ub uncovered terms related to neuronal function (Fig 7G-7I), while peak regions present in all three tissues resided near genes whose molecular functions related to DNA binding and transcription (Fig 7J).

After finding that H2Aub contributes to brain-specific patterns of both gene activation and gene repression, we asked whether H2Aub in the brain was enriched at sites harboring H3K27me3 or H3K27ac. We predicted that, relative to H2A9ub in non-neural organs, H2Aub detected by CUT&RUN in the cerebellum would be enriched within both facultative heterochromatin (H3K27me3) and active enhancers (H3K27ac). After identifying peaks in CUT&RUN data for H3K27ac and H3K27me3 in cerebellum, liver, and kidney, we generated aggregate heatmaps of H2KAub data from all three tissues. H2Aub was more abundant in heatmaps centered around peak summits for H3K27ac and H3K27me3 in the cerebellum than in the liver or kidney (S12K-S12P Fig). H2Aub thus contributes to tissue-specific chromatin states within both activated and repressed loci.

### PRC2-mediated H3K27me1 is extensively remodeled within euchromatic gene bodies

During cerebellar development, dynamic regulation of PRC2-dependent facultative heterochromatin-associated H3K27me3 drives changes in gene expression programs required for neuronal maturation [80]. PRC2-dependent H3K27me1, however, has been explored much less than H3K27me3 in the mammalian brain. To determine whether PRC2-dependent chromatin modifications were, like PRC1-dependent H2Aub, associated with euchromatin in the brain, we began by examining H3K27me1 in the mature 3-month old mouse cerebellum. Here we observed a striking alternation of mutually exclusive euchromatic and heterochromatic zones harboring H3K27me1 and H3K27me3, respectively, each of which often encompassed multiple genes (Fig 8A). Similar non-overlapping H3K27me1 and H3K27me3 domains have been previously shown in embryonic stem cells [32,77,81], suggesting that this arrangement is a fundamental aspect of genome organization. In the brain, regions harboring H3K27me1 were expressed and contained H3K4me3 in promoter regions, whereas heterochromatic regions tended to bear H3K27me3 in the absence of H3K27me1, H3K4me3 and transcriptional activity (Fig 8A). These results indicate that PRC2 acts throughout both the facultative heterochromatin and euchromatin compartments.

Given the localization of H3K27me1, we wondered about the relationship between H3K27me1 and transcript abundance. To prioritize the examination of transcription by excluding cytoplasmic mRNA in RNAseq experiments, we input RNA extracted from nuclei isolated from adult cerebellum. Although H3K27me1 appeared at expressed loci, its abundance seemed to be inversely correlated with that of the corresponding transcript. For instance, highly expressed genes

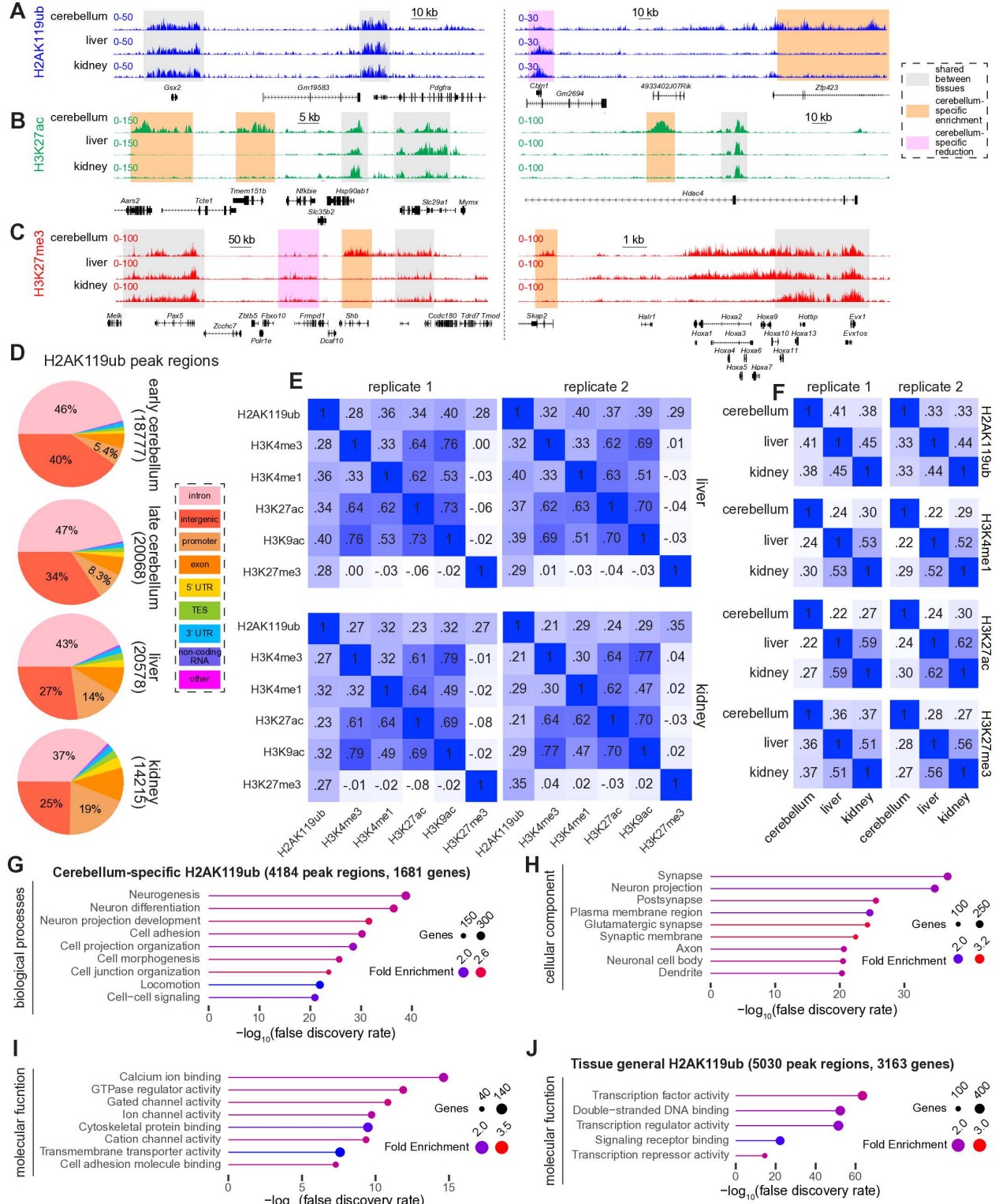

**Fig 7. H2AK119ub has both shared and cerebellum-specific genomic distribution. (A-C)** CUT&RUN tracks of H2AK119ub, H3K27ac, and H3K27me3 in adult cerebellum, liver, and kidney. **(D)** HOMER annotation of H2AK119ub peak regions. **(E)** Pearson correlation matrices of histone modifications in the liver and kidney. **(F)** Pearson correlation matrices comparing histone modifications within neural (cerebellum) and non-neural

(liver and kidney) tissues. **(G)** ShinyGO [79] analysis of biological processes linked to cerebellum-specific H2AK119ub peak regions. **(H)** ShinyGO analysis of cellular components linked to cerebellum-specific H2AK119ub peak regions. **(I)** ShinyGO analysis of molecular functions linked to cerebellum-specific H2AK119ub peak regions. **(J)** ShinyGO analysis of molecular functions linked to H2AK119ub peak regions shared across the adult cerebellum, liver, and kidney.

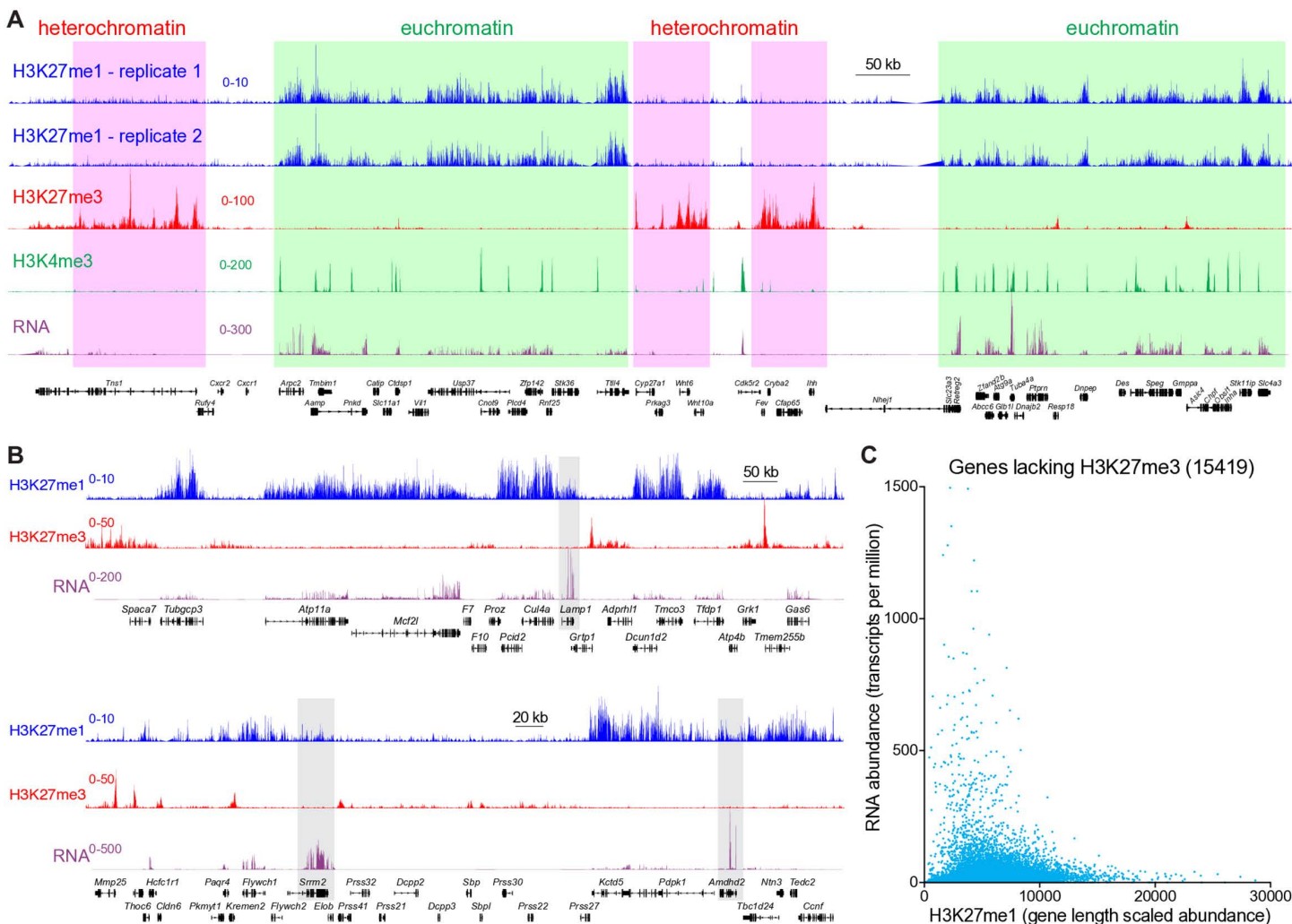

**Fig 8. PRC2-mediated H3K27me1 is enriched within neuronal euchromatin but uncorrelated with transcript abundance. (A)** CUT&RUN and RNAseq tracks from the late cerebellum. Shading regions highlight alternating euchromatic and heterochromatic zones. **(B)** CUT&RUN and RNAseq tracks from the late cerebellum. Shading highlights highly expressed genes with low levels of H3K27me1. **(C)** Scatterplot comparing the gene length-normalized H3K27me1 abundance, as detected by CUT&RUN, to transcript abundance (exons only) for protein-coding genes. Genes whose promoters harbored H3K27me3 were excluded prior to this analysis.

exhibited less H3K27me1 than the surrounding less-abundant transcripts (Fig 8B). After excluding 5560 protein-coding genes whose promoters harbored H3K27me3, plots comparing the gene length-scaled abundance of H3K27me1 to the length-scaled abundance of the corresponding mRNAs failed to reveal any correlation between these readouts, with highly expressed genes tending to harbor some but not too much H3K27me1 (Fig 8C).

To determine whether the pattern of euchromatin-associated H3K27me1 changed over the course of neurodevelopment, we visualized locus-level H3K27me1 CUT&RUN data from the early and late cerebellum. H3K27me1 was lost at some loci and gained at others (Fig 9A), but the real change was in the pattern of distribution: in the early cerebellum H3K27me1 was broadly and uniformly dispersed within both intragenic and intergenic regions, but in the adult cerebellum H3K27me1 became enriched within specific gene bodies. Accordingly, the fraction of H3K27me1 peak summits found within intergenic regions fell over neurodevelopment while the intragenic fraction rose (Fig 9B). Examination of this redistributive process through scatterplots depicting the gene length-normalized abundance of H3K27me1 in the early versus late cerebellum showed a group of genes that gained significant amounts of H3K27me1 by the late timepoint while many other genes lost the modest H3K27me1 levels they had possessed (Fig 9C). Plotting the average of the H3K27me1 signal within gene bodies at the early and late timepoints on the x-axis and the ratio of H3K27me1 late/early on the y-axis segregated these two groups of genes (S13A Fig). Similarly, in metagene heatmaps, H3K27me1 went from being evenly distributed throughout the gene bodies of protein-coding genes in the early cerebellum to become enriched within a minority of protein-coding genes at the late timepoint while being simultaneously depleted from most other genes (Fig 9D). We then sorted loci into deciles according to the gene-length scaled abundance of H3K27me1 as measured in the late cerebellum. In heatmaps depicting the top decile of genes, H3K27me1 was more abundant in the late cerebellum (Fig 9E). Lower deciles showed a decline in the abundance of H3K27me1 by the late timepoint, with the bottom decile showing no detectable H3K27me1. GO analysis of the 526 genes that contained more than twice as much H3K27me1 at the late timepoint demonstrated a striking enrichment for terms related to RNA processing and chromatin (S13B Fig). *Drosha*, which encodes an essential micro-RNA ribonuclease whose dysfunction is implicated in several diseases [82], ranked among the top genes for which H3K27me1 abundance increased over the course of neurodevelopment while being depleted from the flanking regions (S13C Fig). To determine whether loci harboring H2K27me1 in the early cerebellum went on to become trimethylated, we also plotted H3K27me3 data within the deciles previously defined for H3K27me1 (Fig 9F). There was no obvious relationship between the developmental trajectories of H3K27me1 and H3K27me3 within these clusters. Loci in the bottom decile for H3K27me1 harbored the most H3K27me3, in agreement with the mutually exclusive localization of these modifications, but this group of loci did not appear to gain H3K27me3 between the early and late timepoints. These observations suggest that H3K27me3 and H3K27me1 are regulated separately by PRC2 at the level of individual genes. Plotting mRNA abundance within these deciles showed no obvious changes over cerebellar development (Fig 9G). Interestingly, mRNA levels were highest in the 50–60% decile, indicating that genes possessing a small amount of H3K27me1 end up as the most highly transcribed.

Finally, we asked which of the PRC2-dependent histone PTMs—euchromatin-associated H3K27me1 or heterochromatin-associated H3K27me3—varied more across cerebellar development. Pearson correlation matrices using CUT&RUN data as input showed that H3K27me1 and me3 were anti-correlated at both the early and late timepoints in the mouse cerebellum, in agreement with the non-overlapping distribution of these modifications (Fig 9H). Interestingly, H3K27me3 was much more highly correlated between the two timepoints (coefficients of 0.78 and 0.8) than H3K27me1 (coefficients of 0.27 and 0.29). Although production of H3K27me3 in neurons [43] and embryonic stem cells [81] requires both of PRC2's histone methyltransferase subunits, EZH1 and EZH2, differentiating neurons reduce their expression of EZH2 as they exit the cell cycle while continuing to express EZH1 [83], which can monomethylate Lys27 to generate H3K27me1 but does not readily trimethylate it [84–86]. RNAseq analysis of early and late mouse cerebellum showed that *Ezh1* was more highly expressed in late cerebellum than was *Ezh2*, whose product predominated in the early cerebellum (S13D-S13G Fig). These results indicate that heterochromatic EZH2-dependent H3K27me3 is established early in cerebellar development and comparatively maintained, whereas euchromatic EZH1-dependent H3K27me1 is remodeled to a greater degree over a longer period.

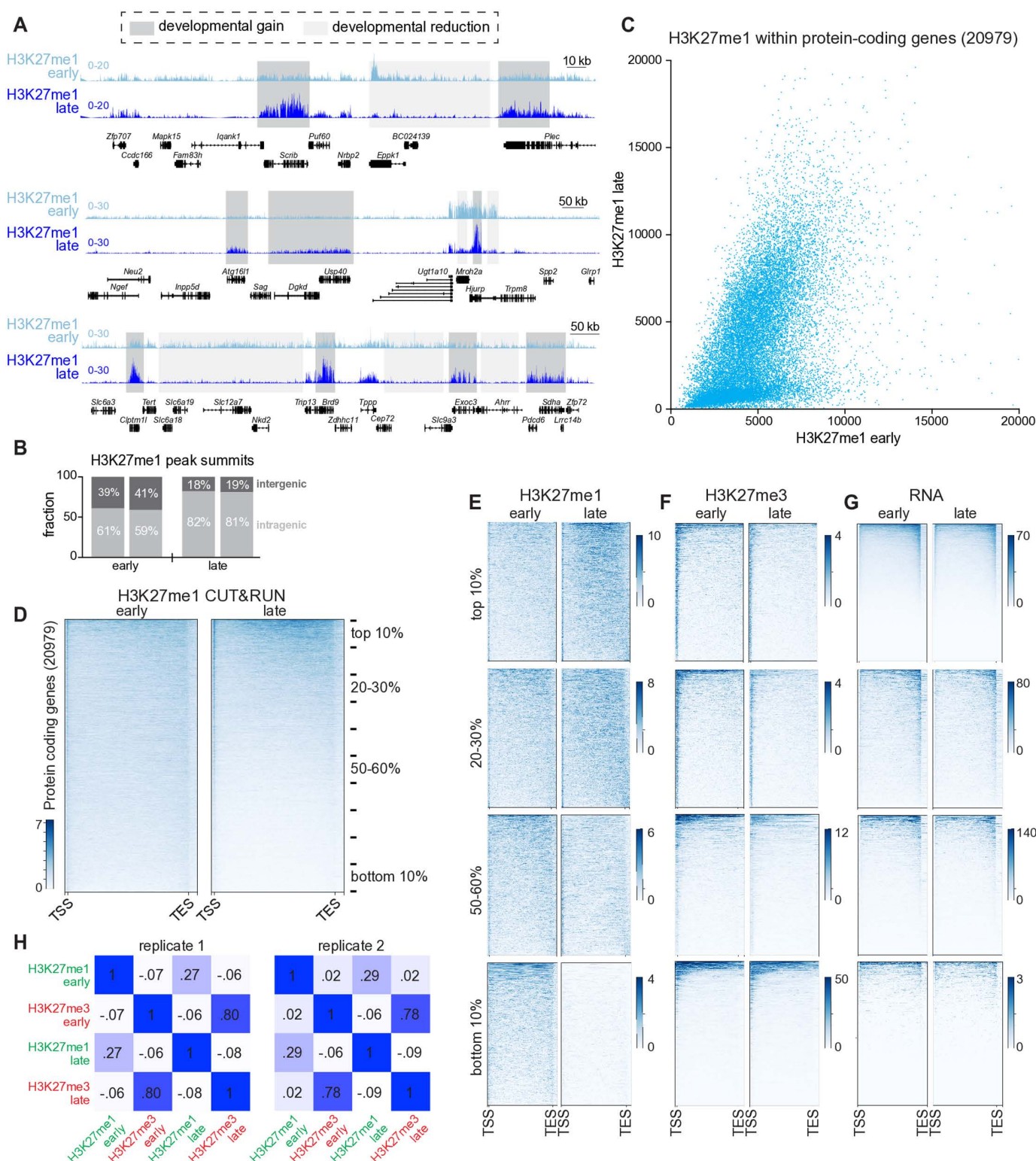

**Fig 9. Neurodevelopmental enrichment of H3K27me1 within gene bodies. (A)** Normalized H3K27me1 CUT&RUN tracks in the early and late cerebellum. **(B)** Stacked bar charts showing the percentage of H3K27me1 peak summits identified by the MACS2 narrow algorithm that resided within intragenic or intergenic regions, as defined by HOMER in the early and late cerebellum (n = 2). **(C)** Scatter plot comparing the gene length-scaled

abundance of H3K27me1 within protein-coding genes in the early versus late mouse cerebellum. **(D)** Metagene heatmaps depicting H3K27me1 CUT&RUN data across protein-coding genes in the early versus late cerebellum. Genes were sorted into deciles according to their length-scaled abundance of H3K27me1 in the late cerebellum. **(E)** Metagene plots of H3K27me1 CUT&RUN data from early and late cerebellum across selected deciles. **(F)** H3K27me3 CUT&RUN data within deciles. **(G)** RNAseq data within deciles. **(H)** Pearson correlation matrices comparing H3K27me1 and H3K27me3 CUT&RUN data from the early and late cerebellum.

## Discussion

Polycomb Repressive Complexes are highly dynamic during neurodevelopment, with PRC1-dependent H2Aub becoming depleted from heterochromatin and enriched in euchromatin, and PRC2-dependent H3K27me1 becoming enriched within transcribed gene bodies. While PRC activity within expressed loci has been described in flies, cancer cells and embryonic stem cells [87–90], PRC activity within euchromatin has never been explored in the context of mammalian neurodevelopment. Genome-wide analyses indicate that loci containing H2Aub within H3K27me3-positive facultative heterochromatin are relatively static in comparison to loci in which H2Aub is unassociated with H3K27me3. In fact, H3K4me3, H3K27ac and H2Aub itself were all more dynamic within the former class. Interestingly, H3K27me3 and H2Aub changed in opposite directions within the former class of loci, indicating PRC1 and PRC2 operate independently as the cerebellum matures. The euchromatin-associated modification H3K27ac also behaved differently at sites harboring H2Aub alone, becoming enriched over time, while H3K27ac was depleted from loci that harbored both H2Aub and H3K27me3. Furthermore, the relationship between H2Aub and H3K27ac also changed over time, as early active enhancers that gained H2Aub tended lose H3K27ac, whereas late active enhancers tended to gain both H2Aub and H3K27ac. H3K4me3-positive promoters also gained both H2Aub and H3K4me3 during neurodevelopment. The observation that activating modifications move in concert with H2Aub over neurodevelopment is one of the most surprising and important results in this study.

Because differential activation and silencing of cis-regulatory elements gives rise to tissue-specific gene expression programs [6,91], our identification of brain-specific patterns of enhancer and promoter ubiquitination carries implications for the role of ubiquitin signaling in neuronal specification and maintenance of neuronal identity. Dynamic modulation of chromatin state at enhancers underlies an array of essential neuronal processes, such as driving long-term changes in gene expression associated with experience-dependent plasticity [6,7,10]. Our findings thus provide a useful resource for interpreting studies of typical neurodevelopment as well as disorders of neurodevelopment caused by mutations in PRC components and their interactors [39–41,48,92,93]. We found that H2AK119ub downregulates the activity of both typical H3K4me3-only promoters and bivalent promoters, in line with the repressive function of PRCs. And although we do not directly demonstrate this, ubiquitination is also likely to temper the activity of active enhancers. Ubiquitination could promote the removal of euchromatin-associated modifications from the early cerebellum, as we observed for H3K27ac. Ubiquitination could also influence cis regulation via the incorporation histone variants such as H2A.z, which is known to be ubiquitinated [94], enriched at polycomb target genes [95], and deposited at promotors in response to neural activity in the cerebellum [96]. It is also possible that H2Aub influences the activity of cis-regulatory elements by modulating chromatin compaction, although this remains to be tested experimentally [45,97,98]. Ubiquitination influences the tendency of its target proteins to undergo phase separation [99,100], a process that contributes the formation of polycomb bodies [101–104], but recent studies have shown that the influence of H2Aub on chromatin condensation is context-specific, with contributions to both activation and repression [4,105]. Therefore, in addition to recruiting its effectors to chromatin, H2Aub likely modulates the biophysical properties of cis-regulatory elements.

Our data also showed that PRC2-dependent neuronal H3K27me1 was highly abundant across broad euchromatic regions while being excluded from heterochromatin. Although a recent study linked H3K27me1 deposition to stress response in D1 medium spiny neurons in the nucleus accumbens [106], PRC2 function in the brain has been studied primarily in relation to H3K27me3 and facultative heterochromatin [3]. The arrangement of H3K27me1 and H3K27me3 into adjacent non-overlapping domains that we observed establishes that PRC2 modifies essentially the entire protein-coding

neuronal epigenome. But whereas H3K27me3 is established in the developing brain and thereafter relatively maintained, acquisition of the mature pattern of H3K27me1 takes much longer. What, then, is the function of neuronal H3K27me1? H3K27me1 could represent a form of transcriptional memory to promote transcriptional fidelity, as has been previously proposed [77]. Another possibility, not mutually exclusive, would be that H3K27me1 functions as a less repressive version of H3K27me3. This would be compatible with our observation that H3K27me1 was uncorrelated or even anti-correlated with transcript abundance. If this is the case, the relationship between H3K27me1 and H3K27me3 would represent the repressive analog of the relationship between H3K4me1 and H3K4me3 in gene expression, in which the former (H3K4me1) recruits transcriptional machinery less avidly than the latter (H3K4me3) [107,108].

In conclusion, PRC-dependent histone modifications show considerable remodeling over the course of neurodevelopment, particularly within the unexpected context of euchromatin. These observations lay the groundwork for future studies into ubiquitin-mediated chromatin regulation in the brain, with implications for our understanding of the molecular control of cell-intrinsic mechanisms of neurodevelopment and the pathogenesis of neurologic disease.

## Materials & methods

### Ethics statement

Animals were cared for in accordance with NIH guidelines. All experimental methods were approved by the UCSD Institutional Committee on the Use and Care of Animals under the protocol number S20121.

### Resource availability

*Lead contact:* Further information and requests for resources and/or reagents should be directed to the Lead Contact, Cole Ferguson (cferguson@health.ucsd.edu).

*Materials availability:* Any other non-commercially available reagent or input biomaterial generated or used during this study will be made freely available to interested researchers.

### Experimental models

**Mice.**  Sex matched-littermate controls on the Jackson Labs C57BL/6 background were used in all experiments, and the ages at which animals were used is reported in figure legends. Animals were housed in a 12:12 light:dark cycle. Biological replicates were sex-matched littermates, and both male and female animals were examined in the course of the major experiments. None of the main findings in this work varied by sex in the tissues we examined.

### Method details

**Nuclear isolation.**  Mice were deeply anesthetized with isofluorane before decapitation and extraction of the while cerebellum, liver and/or kidney. The liver and kidney were further dissected, before choosing the identical lobe in the case of the liver and pole in the case of the kidney, inputting ~100 mg of tissue. Tissue was finely minced in lysis buffer (Nuclei EZ Lysis Buffer supplemented with 1x Halt combined protease and phosphatase Inhibitor cocktail, 10 mM sodium butyrate and 1.5 mM iodoacetamide) prior to mechanical homogenization with Dounce A pestle in 2 mL of Lysis Buffer. This suspension was incubated on ice for 5 minutes before centrifugation at 500 x g for 5 minutes, with acceleration and deceleration on the lowest setting, at 4 degrees. The supernatant was aspirated and nuclei were resuspended in EZ lysis buffer and incubated on ice for an additional 5 minutes prior to centrifugation. Examination of nuclei under DIC demonstrated abundant intact nuclei for all the tissues examined. Intact nuclei were nevertheless stained by trypan blue, indicating permeabilization by this procedure. Nuclei were resuspended in CUT&RUN wash buffer and filtered through a 40 μm mesh. This process routinely yielded ~6 million nuclei from a single mouse cerebellum, liver and kidney samples.

**Native CUT&RUN.** For CUT&RUN performed under native conditions, nuclei were resuspended in wash buffer (20 mM HEPES pH 7.5, 150 mM NaCl, 0.5 mM spermidine supplemented with Roche complete EDTA-free protease inhibitor tablet). Pelleted nuclei were resuspended twice in wash buffer and subjected to centrifugation at 500 x g for 5 minutes. Nuclei were resuspended in 1 mL, filtered by gravity through a 40 μm mesh, and counted with a Countess automated cell counter (Invitrogen). In general, between 250,000–500,000 nuclei were used in each CUT&RUN experiment, and the cellular input was kept the same across all replicates. Nuclei were bound to Concanavalin A-coated magnetic beads. Following bead binding, the supernatant was discarded, and the bead-nuclei slurry was resuspended in 75 uL of antibody buffer (wash buffer supplemented with 0.005% digitonin and 2 mM EDTA) containing primary antibodies at a concentration of 1:50. Antibody incubation was carried out overnight at 4°C on a nutator. Samples were washed with cold cell permeabilization buffer (Wash buffer with 0.005% digitonin) to remove unbound antibody and then incubated with 1x Protein A/G-MNase in cell permeabilization buffer for 15 minutes at room temperature on a shaker. Samples were washed with cold cell permeabilization buffer and then resuspended in 50 μL cell permeabilization buffer containing 2 mM $CaCl_2$. Samples were digested for 2 hours at 4°C on a nutator. The digestion reactions were quenched with 33 μL of STOP buffer (340 mM NaCl, 20 mM EDTA pH 8, 4 mM EGTA pH 7.7, RNase A 0.05 mg/mL, Glycogen 0.05 mg/mL). Samples were then incubated for 10 minutes at 37°C to release digested chromatin fragments. The supernatant containing CUT&RUN fragments was transferred to fresh tubes and DNA was purified via column purification.

**Fixed CUT&RUN.** CUT&RUN detection of SUZ12 and EZH2 under cross-linked conditions was carried out using a similar protocol overall, but with important differences. After tissue homogenization, the supernatant was removed and nuclei were briefly fixed in a 1% formaldehyde in PBS for 1 minute at room temperature before quenching in 125 mM glycine. Subsequently, wash, antibody and permeabilization buffers were supplemented with 1% Triton X-100 and 0.05% SDS. After stopping the chromatin digestions and retrieving liberated chromatin complexes, decrosslinking was performed 0.8 μL of 10% SDS and 1 μL of 20 μg/μL of Proteinase K were added to each sample before incubation at 55°C overnight.

## Evaluation of antibody specificity

The specificity of several antibodies used in this study could be directly evaluated using the CUTANA K-MetStat panel, which includes modified nucleosomes bound to magnetic beads, each harboring a specific barcode sequence. This collection of modified nucleosomes was then added to CUT&RUN experiments. After library preparation and sequencing, the specificity of each antibody could be computed by comparing the abundance of the sequences associated with each modified nucleosome via the supplier's provided program and analysis spreadsheet.

## CUT&RUN library preparation and sequencing

Library preparation for CUT&RUN experiments was performed according to the manufacturer's specifications using SPRIselect beads. The concentration of DNA yielded from CUT&RUN was measured on a Qubit device and between 25 and 50 ng of input DNA was used for library preparation, with the amount of input DNA kept constant for all samples using the same antibody. 14 PCR cycles were used in the final amplification with Illumina-compatible adaptors, universal i5 primer and barcoded i7 primers. After measuring the concentration of the amplified product on Nanodrop, all samples were diluted to the same concentration for a given antibody, ranging from 10 ng/uL to 40 ng/uL. Tapestation was performed to determine the precise concentration and fragment distribution range and to ensure there were not significant amount adaptor dimers. Samples were pooled for sequencing 100 bp reads in paired-end configuration on an Illumina NovaSeqX platform at the UCSD Institute for Genomic Medicine. We aimed for >25 million reads for all samples except for H3K4me3, for which 15 million reads were adequate given the narrow distribution of this histone modification.

## CUT&RUN data processing

We used FastQC to evaluate read quality, including base quality scores and adapter contamination. Compressed sequenced reads in fastq.gz format were first trimmed using Trimmomatic v.0.39 [109]. We then aligned samples to the mm10 genome using Bowtie2 v.2.3.4.1 [110] and sorted BAM files, removing unmapped fragments using SAMtools v.1.14 [111]. During alignment we set a minimum Phred score of 33 and employed the --dovetail option. We then converted alignment files to.bigwig files using the bamCoverge tool in deepTools v.3.5.0 [26] with 50 bp bins. Duplicate reads were not removed. Peak calling was performed using SEACR (Sparse Enrichment Analysis for CUT&RUN) v.1.1 [27] and MACS2 (Model-Based Analysis of ChIP-seq) [59] in broad and narrow formats. Samples of different replicates collected in the same experiment were normalized to each other to correct for different loading between samples..bigwig and.bed files were examined in Integrated Genome Viewer and visualized using KaryoplotR v.1.22.0 [112]. We obtained regional overlap between different samples using the 'intersect' functionality of BEDTools v.2.29.2 [28].

## Normalization of CUT&RUN data

After parsing the genome into 50 bp bins, we identified local maxima and generated a blacklist of 271 intergenic peaks representing spurious alignment artifacts. We quantified the height of the leftover authentic peaks and recorded the value of the height for the 99th percentile peak recorded. The ratio of the values of the 99th percentile peaks between samples served as a scaling factor by which the height of every bin in one sample could be multiplied to normalize samples to one another.

## DiffBind analysis

Differential peak analysis was performed on SEACR-called peaks using DiffBind (v. 3.14) [65] with default parameters (summit width = 400 bp). Significantly differential peaks were identified using DESeq2 [25] within Diffbind and only peaks with FDR < 0.05 were used for further analysis. Functional annotation of significant peaks was performed using the "annotatePeak" function of ChIPseeker (v.1.4) [113] using the TxDb.Mmusculus.UCSC.mm10.knownGene (v. 3.10) database.

## Heatmaps of signal grouped by developmental H2AK119ub dynamics

To visualize chromatin state changes across distinct categories of H2AK119ub regulation, peak regions were grouped based on differential analysis of H2AK119ub signal using the DiffBind R package. Peaks were categorized as no change, significantly increased, or significantly decreased over neurodevelopment (adjusted p < 0.05, fold change > 1). For each group, CUT&RUN signal for H2AK119ub, H3K27me3, H3K4me3, H3K27ac, and H3K9ac across both timepoints was extracted from bigWig files using computeMatrix with the reference-point setting. plotHeatmap was then used to visualize the signal centered around peak summits and sorted by decreasing histone modification abundance.

## Genome-wide aggregate heatmaps

Genome-wide heatmaps were generated using deepTools v.3.5. Plots centered around peak summits were produced in reference to.bed files output from peak calling analysis of CUT&RUN data from mouse tissue. computeMatrix was used to represent each locus from bigwig CUT&RUN data, where the rows were loci and the columns were 50 bp bins. Matrices were subsequently plotted as heatmaps using the plotHeatmap tool. For metagene plots, in which the signal in bigwig tracks was scaled to gene length, the metagene flag was used during computeMatrix. plotHeatmap was then used to visualize the abundance of different histone PTMs.

## Heatmaps within PRC1-Only (H2AK119ub alone) and PRC1–PRC2 (H2AK119ub & H3K27me3) loci

To examine histone modification profiles associated with distinct Polycomb modules, H2AK119ub and H3K27me3 peak regions were intersected at each timepoint using BEDTools to define co-occupied (PRC1–PRC2) and H2AK119ub-only

(PRC1-only) regions. Additional sets of H2AK119ub-only regions unique to the early or late cerebellum were similarly defined. For each group, CUT&RUN signal for H2AK119ub, H3K27me3, H3K4me3, H3K27ac, and H3K9ac at early and late timepoints was extracted using computeMatrix with the reference-point setting centered on peak summits. Signal intensity was visualized using plotHeatmap sorted by decreasing signal abundance.

### Length-scaled signal abundance of CUT&RUN data

To analyze the signal from CUT&RUN bigwig tracks in a manner that scaled the locus-specific signal to the length of each locus within.bed file-defined regions, we employed the computeMatrix tool and the scale-regions flag to scale all regions defined by the.bed file to a standardized length. To generate quantitative length-scaled data for scatter plots, the signal for each locus was summed across rows, representing the total scaled signal for a given histone modification.

### Correlation matrices

Pearson correlations coefficients between different experiments were generated using scipy v.1.5.3. Input data files were first binned into 50 base pair intervals and alignment artifacts were removed. For region-specific analysis within specific chromatin compartments identified by calling peak regions in CUT&RUN data, genomic regions not included in the peak region.bed file were filtered out before computing the Pearson r values for each combination of experiments as above.

### Functional annotation of H2AK119ub peak regions

Peak regions which were shared between biological replicates were identified using BEDTools. The resulting intersected list of peak regions were assigned genomic annotations using annotatePeaks.pl in Homer v.5.1 [67], with the annStats parameter enabled. A truncated list of the 12 most relevant annotations was used for downstream analysis.

### Differential analysis of CUT&RUN data

Log2-normalized consensus peak counts were analyzed for differential peak abundance using the R BioConductor packages edgeR [71] and limma [72]. The experimental design was modeled upon time point (~0 + time). The lmFit function in limma followed by the eBayes function was used to fit the design on log2-normalized peak counts per region. Significance was defined by using an adjusted p-value cut-off of 0.05 after multiple testing correction using a moderated t-statistic in limma.

### RNAseq

Mouse cerebellar nuclei were isolated using the same nuclear isolation protocol for CUT&RUN. Isolated nuclei were homogenized in Trizol. Total nuclear RNA was isolated using the column-based Qiagen RNeasy kit. Library prep was performed using 1 ug of input RNA with the Illumina Stranded Total RNA Prep Ligation with Ribo-Zero Plus kit. After depleting ribosomal RNA and performing cDNA synthesis, the DNA was quantitated using Qubit and equalized across samples. Illumina-compatible dual index primers were used during 15 PCR cycles. The prepared libraries were subsequently sequencing using the Illumina NovaSeq S4 platform to generate 100 bp paired-end reads. Each sample was sequenced to a depth of at least 30 million reads. RNAseq libraries were aligned to mm10 using STAR v.2.7.9a. Alignment files were then converted to bigwig format using deepTools bamCoverage v.3.5.0. DESeq2 [25] was then used to compute normalized reads and adjusted P-values.

### Gene ontology analysis

A list of the chromosomal coordinates of all known genes in the mm10 genome along with their Refseq IDs was obtained from the UCSC Genome Browser. The closest RefSeq genes within 10 kb of each.bed file locus was obtained using

BEDTools v.2.29.2. The gene IDs left after this filtering was the gene list used as input to ShinyGO v.0.77 [79] for GO analysis with an FDR cutoff of 0.05. No background was inputted.

## Comparison of H3K27me1 and mRNA abundance

To assess the relationship between H3K27me1 abundance and gene expression, we excluded genes likely to be repressed by H3K27me3. H3K27me3 peak regions were defined using SEACR peak calling on CUT&RUN data from the adult cerebellum. Protein-coding gene annotations (mm10) were obtained via Ensembl BioMart, and promoter regions were defined as 1 kb upstream of the transcription start site. To identify genes with nearby repressive marks, the BEDTools closest function was used to calculate the distance between each gene's promoter region and the nearest H3K27me3 peak. Genes with a H3K27me3 peak within 5 kb upstream of the promoter were excluded from further analysis. For the remaining genes, H3K27me1 abundance across gene bodies was quantified from CUT&RUN bigwig files using deepTools computeMatrix with the scale-regions flag to normalize to gene length. Corresponding RNA abundance was calculated as transcripts per million, as quantitated by DESeq2, including only exons. These values were plotted per gene with gene-length-scaled H3K27me1 signal on the x-axis and TPM on the y-axis.

## Analysis of transcript abundance in genes with H3K4me3-Only or bivalent promoters stratified by H2AK119ub dynamics

To assess the impact of promoter-associated H2AK119ub on transcription, we first defined two promoter classes using SEACR-called CUT&RUN peak regions in the early cerebellum: (a) H3K4me3-only promoters by the presence of H3K4me3 peaks without overlapping H3K27me3, and (b) bivalent promoters as overlapping H3K4me3 and H3K27me3 peaks. Promoter-associated H2AK119ub signal was quantitated across both promoter classes using deepTools computeMatrix in scale-region mode. Differential analysis of promoter-associated H2AK119ub signal between early and late cerebellum was performed using limma voom, with adjusted p-value < 0.05 and fold-change > 1.0 defining significant increase, and adjusted p > 0.05 defining no change. bedTools closest was used to assign promoter regions to Ensembl BioMart-derived protein-coding genes for the mm10 reference genome build. Genes associated with each promoter category (H3K4me3-only or bivalent) were grouped by H2AK119ub dynamics (increased vs. unchanged). RNAseq signal from nuclear RNA was plotted in metagene format using computeMatrix with a 2 kb window upstream and downstream of each gene. Heatmaps visualizing early and late mRNA abundance were generated with plotHeatmap for each of the six sets.

## Chromatin hidden Markov model

For each of the histone modifications of interest, we used of ChromHMM v.1.18, which applies a multivariate Hidden Markov Model to infer chromatin states from the CUT&RUN data collected, using data from IgG CUT&RUN as the negative control. Data from two biological replicates at each timepoint were combined. We binarized out input data using the BinarizeBed utility in ChromHMM. The full-stack chromatin state models were then identified using the 'Learn Model', which applies a multivariate Hidden Markov Model to the data. We chose a 15-state model because it most faithfully delineated major chromatin states, including promoters, enhancers, actively transcribed loci and constitutive heterochromatin, as well as other important functionally defined loci. Transition state data generated a diagonal line in the matrix, establishing the validity of downstream emission data.

## Graphing and statistical analysis

All plots were generated in GraphPad Prism10 and modified in Adobe Illustrator 2021. All p-values were computed using the tests indicated in the figure legends in Prism or in R.

## Supporting information

**S1 Table.  Reagents and other tools used in this study.**
(DOCX)

**S2 Table.  Coordinates of blacklisted peak regions.**
(XLSX)

**S3 Table.  Raw numerical data for all graphs.**
(XLSX)

**S1 Fig.  Percentile-based normalization of specific CUT&RUN data.** (A) A panel of nucleosomes modified by specific histone PTMs and harboring modification-specific DNA sequences were spiked-into CUT&RUN experiments in mouse cerebellum using antibodies to H3K4me3, H3K4me1, H3K27me3, and H3K9me3. Nucleosomes modified by acetylation were not present in this panel. The number of reads derived from DNA sequences associated with each spiked-in modified nucleosome was normalized to the total number of reads derived from all the spiked-in modified histones. (B) Overview of pipeline for analyzing CUT&RUN data. (C) Strategy for normalizing CUT&RUN data. (D) Normalized CUT&RUN data for H2AK119ub in two biological replicates. Peaks called using SEACR. (E) Normalized H2AK119ub data across a broader genomic region. (F) Coverage tracks of CUT&RUN data for H3K27me3 detected in the mouse cerebellum, followed by normalization of shallowly sequenced sample B to deeply sequenced sample A. (G) Scatter plot comparison of peak heights in H3K27me3 CUT&RUN data from mouse cerebellum between samples A and B before and after normalization.
(TIF)

**S2 Fig.  Concordance of normalized H2AK119ub CUT&RUN data across timepoints and replicates.** (A-C) Genome browser tracks showing normalized H2AK119ub CUT&RUN signal across four biological replicates from early (P12) and late (3 month) mouse cerebellum.
(TIF)

**S3 Fig.  H2AK119ub colocalizes with PRC2 components at repressed loci.** (A-D) CUT&RUN detection of H2AK119ub, H3K27me3, EZH2, and SUZ12 in the early cerebellum at repressed loci encoding transcription factors.
(TIF)

**S4 Fig.  Repression of early-expressed genes through coordinated PRC1 and PRC2 activity accompanies.** (A) CUT&RUN detection of the indicated histone modifications at the *Insm1* locus in early and late cerebellum shown alongside RNAseq tracks. (B) CUT&RUN and RNAseq tracks at the *Mtss1* locus. (C) Normalized counts for RNAseq detection of *Insm1* transcripts in nuclei isolated from early and late cerebellum (n = 2, adjusted p-value computed by DESeq2). (D) Same as in (C), for *Mtss1*.
(TIF)

**S5 Fig.  H2AK119ub is depleted from facultative heterochromatin, as detected in the late cerebellum.** (A) Heatmaps depicting normalized H2AK119ub CUT&RUN data in the early and late cerebellum centered around peak summits for H3K27me3 detected in the late cerebellum. (B) Volcano plot depicting H2AK119ub abundance, as detected in normalized CUT&RUN data from early and late cerebellum, within H3K27me3 peak regions detected in the late cerebellum. The significance threshold was an adjusted p-value of <0.05, as computed by edgeR and Limma (n = 4). (C) $Log_2$-$log_2$ scatterplot comparing the ratio of H2AK119ub (late/early) to the ratio of H3K27me3 (late/early) within H3K27me3 peak regions identified in the early cerebellum (n = 4 for H2AK119ub, n = 2 for H3K27me3).
(TIF)

**S6 Fig. H2AK119ub and H3K4me3 become coordinately enriched within H3K4me3-positive promoters during cerebellar neurodevelopment.** (A) Heatmaps depicting normalized H2AK119ub CUT&RUN data in the early and late cerebellum centered around peak summits identified by MACS2 narrow for H3K4me3 in the late cerebellum. (B) Volcano plot depicting H2AK119ub abundance, as detected in normalized CUT&RUN data from early and late cerebellum, within H3K4me3 peak regions detected in the late cerebellum. The significance threshold was an adjusted p-value of <0.05, as computed by edgeR and Limma (n = 4). (C) Log$_2$-log$_2$ scatterplot comparing the ratio of H2AK119ub (late/early) to the ratio of H3K4me3 (late/early) within H3K4me3 peak regions identified in the early cerebellum (n = 4 for H2AK119ub, n = 2 for H3K4me3). (D) Log$_2$-log$_2$ scatterplot comparing the ratio of H2AK119ub (late/early) to the ratio of H3K4me3 (late/early) within H3K4me3 peak regions identified in the late cerebellum (n = 4 for H2AK119ub, n = 2 for H3K4me3).
(TIF)

**S7 Fig. Neurodevelopmental enrichment of H2AK119ub within CpG islands.** (A) CUT&RUN detection of H2AK119ub in the cerebellum within the *Bcat1* locus harboring promoter-associated CpG islands. Shaded regions highlight CpG islands exhibiting neurodevelopmental increase in H2AK119ub. (B) Same as (A), showing neurodevelopmental decrease in H2AK119ub at the *Isl1* locus. (L) Violin plots depicting the length-scaled abundance of H2AK119ub in the early and late cerebellum within CpG islands. P-value and D statistic by K-S D test (n = 4). (C) Fraction of H2AK119ub peak regions that overlap with CpG islands in the early and late cerebellum. The two middle percentages reflect: the fraction of CpG islands that overlap with H2AK119ub peak regions and the fraction of H2AK119ub peak regions that overlap with CpG islands. (D) Heatmaps depicting normalized H2AK119ub CUT&RUN data in the early and late cerebellum centered around CpG islands.
(TIF)

**S8 Fig. H2AK119ub undergoes limited developmental enrichment within H3K4me1-positive cis-regulatory elements.** (A) Violin plots showing length-scaled H2AK119ub abundance within H3K4me1 peak regions detected in the early and late cerebellum. P-value and D statistic by K-S D test (n = 4). (B) Heatmaps depicting normalized H2AK119ub CUT&RUN data in the early and late cerebellum centered around H3K4me1 peak summits identified by the MACS2 narrow algorithm in the early cerebellum. (C) Same as (B), shown for peak summits detected in the late cerebellum.
(TIF)

**S9 Fig. H2AK119ub is developmentally enriched at H3K27ac-positive enhancers in the late cerebellum.** (A) Heatmaps depicting normalized H2AK119ub CUT&RUN data centered around H3K27ac peak summits identified by MACS2 narrow in the late cerebellum. (B) Heatmaps depicting normalized H2AK119ub CUT&RUN data centered around active enhancers (intersection of SEACR-defined H3K27ac peaks and MACS2 broad H3K4me1 peaks) detected in the late cerebellum. (C) Volcano plot depicting H2AK119ub abundance, as detected in normalized CUT&RUN data from early and late cerebellum, within H3K27ac peak regions detected in the late cerebellum. The significance threshold was an adjusted p-value of <0.05, as computed by edgeR and (n = 4). (D) CUT&RUN and RNAseq tracks showing an active enhancer cluster with neurodevelopmental gain of H2AK119ub. (E) CUT&RUN and RNAseq tracks showing an active enhancer cluster with neurodevelopmental gain of H2AK119ub and H3K27ac.
(TIF)

**S10 Fig. H2AK119ub is neurodevelopmentally enriched within H3K9ac-positive euchromatin.** (A) Violin plots showing length-scaled H2AK119ub abundance within H3K9ac peak regions detected in the early and late cerebellum. P-value and D statistic by K-S test (n = 4). (B) Heatmaps depicting H2AK119ub CUT&RUN data centered on MACS narrow-defined H3K9ac peak summits from early cerebellum. (C) Heatmaps depicting H2AK119ub CUT&RUN data centered on H3K9ac peak summits from late cerebellum. (D) Volcano plot depicting H2AK119ub abundance, as detected in normalized CUT&RUN data from early and late cerebellum, within H3K27ac peak regions detected in the early cerebellum. The

significance threshold was an adjusted p-value of <0.05, as computed by edgeR and Limma (n = 4). (E) Volcano plot depicting H2AK119ub abundance, as detected in normalized CUT&RUN data from early and late cerebellum, within H3K27ac peak regions detected in the late cerebellum. The significance threshold was an adjusted p-value of <0.05, as computed by edgeR and Limma (n = 4). (F) Heatmaps depicting H3K9ac CUT&RUN data within loci with no change, increased or decreased H2AK119ub over neurodevelopment. (G) Heatmaps depicting H3K9ac CUT&RUN data within loci harboring both H2AK119ub and H3K27me3. (H) Heatmaps depicting H3K9ac CUT&RUN data within loci harboring H2AK119ub in the absence of H3K27me3.
(TIF)

**S11 Fig. Increasing overlap between H2AK119ub peak regions and activating modifications in the late cerebellum.** Fraction of SEACR-defined H2AK119ub peak regions that overlap with peak regions for different histone modifications detected in the early and late cerebellum. Peak regions were called using the SEACR algorithm for all modifications except H3K4me1, when MACS2 broad was used. Inputted peak regions reflect the consensus between two replicates, defined by intersecting overlapping regions found in both. Percentages at the top reflect the fraction of peak regions for each modification that overlap with H2AK119ub peak regions. Bottom percentages reflect the fraction of H2AK119ub peak regions that overlap with peak regions for the comparison modification.
(TIF)

**S12 Fig. H2AK119ub has tissue-specific distributions at active enhancers and silencers.** (A-C) Venn diagrams depicting the overlap between SEACR-called peak regions for H2AK119ub CUT&RUN in the cerebellum and liver (A), cerebellum and kidney (B), and liver and kidney (C). (D-F) Same as (A-C), for H3K27ac. (G-I) Same as (A-C), for H3K27me3. (J) Bar graphs depicting the fraction of unique peak regions. (K–P) Heatmaps of H2AK119ub in cerebellum, liver, and kidney centered on H3K27ac or H3K27me3 peak summits called in cerebellum (K–L), liver (M–N), or kidney (O–P).
(TIF)

**S13 Fig. H3K27me1 is extensively redistributed across gene bodies over neurodevelopment.** (A) Minus-average plot comparing the length-scaled bundance of H3K27me1 across all protein-coding genes. (B) ShinyGO analysis of 526 genes exhibiting > 2-fold increase in H3K27me1 over neurodevelopment. (C) H3K27me1 CUT&RUN tracks at the *Drosha* locus. (D) RNAseq tracks at the *Ezh1* locus in early and late cerebellum. (E) Bar graph comparing DESeq2 normalized *Ezh1* mRNA counts in early and late cerebellum. (F) Same as (D), at the *Ezh2* locus. (G) Same as (E), at the *Ezh2* locus.
(TIF)

## Acknowledgments

We thank Vicky L. Brandt for many insightful discussions and a commitment to substance and clarity. We thank members of the Ferguson lab and Joseph Corbo for their thoughtful feedback on this study. We also thank Kristen Jepsen and the UCSD IGM. This publication includes data generated at the UC San Diego IGM Genomics Center utilizing an Illumina NovaSeq X Plus that was purchased with funding from a National Institutes of Health SIG grant (#S10 OD026929).

## Author contributions

**Conceptualization:** Aditya Parmar, Anjali Srinivasan, Leya Ledvin, Cole J. Ferguson.

**Data curation:** Aditya Parmar, Anjali Srinivasan, Abijith Augustine, Owin Gong, Addison C. Bullard, Riya Kalra, Dylan Pilz, Challana E. Tea, Roman Sasik, Kathleen M. Fisch, Cole J. Ferguson.

**Formal analysis:** Aditya Parmar, Anjali Srinivasan, Lena Krockenberger, Abijith Augustine, Owin Gong, Addison C. Bullard, Riya Kalra, Dylan Pilz, Ying Sun, Kathleen M. Fisch.

**Funding acquisition:** Cole J. Ferguson.

**Investigation:** Aditya Parmar, Anjali Srinivasan, Abijith Augustine, Owin Gong, Addison C. Bullard, Leya Ledvin, Dylan Pilz, Jonathan Tawil, Challana E. Tea, Olivia Urso, Larissa M. Kaube, Kathleen M. Fisch, Cole J. Ferguson.

**Methodology:** Aditya Parmar, Anjali Srinivasan, Lena Krockenberger, Abijith Augustine, Owin Gong, Riya Kalra, Leya Ledvin, Dylan Pilz, Challana E. Tea, Kelly C. Wang, Larissa M. Kaube, Ying Sun, Roman Sasik, Kyle J. Gaulton, Kathleen M. Fisch.

**Software:** Aditya Parmar, Anjali Srinivasan, Lena Krockenberger, Abijith Augustine, Owin Gong, Addison C. Bullard, Riya Kalra, Ying Sun, Roman Sasik, Kyle J. Gaulton, Kathleen M. Fisch.

**Supervision:** Leya Ledvin, Olivia Urso, Kyle J. Gaulton, Kathleen M. Fisch, Cole J. Ferguson.

**Validation:** Aditya Parmar, Anjali Srinivasan, Lena Krockenberger, Challana E. Tea, Roman Sasik, Kathleen M. Fisch, Cole J. Ferguson.

**Visualization:** Aditya Parmar, Anjali Srinivasan, Lena Krockenberger, Abijith Augustine, Owin Gong, Addison C. Bullard, Riya Kalra, Leya Ledvin, Dylan Pilz, Challana E. Tea, Kelly C. Wang, Ying Sun, Roman Sasik, Kathleen M. Fisch, Cole J. Ferguson.

**Writing – original draft:** Aditya Parmar, Cole J. Ferguson.

**Writing – review & editing:** Aditya Parmar, Cole J. Ferguson.

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
