## [Decision Letter · Decision Letter 0]

3 Jun 2025

PGENETICS-D-25-00481

Polycomb repressor complexes regulate euchromatin during cerebellar neurodevelopment

PLOS Genetics

Dear Dr. Ferguson,

Thank you for submitting your manuscript to PLOS Genetics. After careful consideration, we feel that it has merit but does not fully meet PLOS Genetics's publication criteria as it currently stands. Therefore, we invite you to submit a revised version of the manuscript that addresses the points raised during the review process.

Please submit your revised manuscript within 30 days Jul 03 2025 11:59PM. If you will need more time than this to complete your revisions, please reply to this message or contact the journal office at plosgenetics@plos.org. Please include the following items when submitting your revised manuscript:

We look forward to receiving your revised manuscript.

Kind regards,

Marisa S Bartolomei

Academic Editor

PLOS Genetics

Marnie Blewitt

Section Editor

PLOS Genetics

Aimée Dudley

Editor-in-Chief

PLOS Genetics

Anne Goriely

Editor-in-Chief

PLOS Genetics

**Additional Editor Comments:**

The reviewers were enthusiastic about the findings but all three have suggested a number of revisions that should help to improve the paper and the strength of the conclusions.

**Journal Requirements:**

At this stage, the following Authors/Authors require contributions: Anjali Srinivasan, Aditya Parmar, lena krockenberger, abijith augustine, Owin Gong, leya ledvin, addison bullard, Kelly Wang, jonathan tawil, riya kalra, dylan pilz, olivia urso, larissa kaube, ying sun, roman sasik, Kyle Gaulton, kathleen fisch, and Cole J. Ferguson. Please ensure that the full contributions of each author are acknowledged in the "Add/Edit/Remove Authors" section of our submission form.

The list of CRediT author contributions may be found here: https://journals.plos.org/plosgenetics/s/authorship#loc-author-contributions

https://journals.plos.org/plosgenetics/s/submission-guidelines#loc-parts-of-a-submission

4) We do not publish any copyright or trademark symbols that usually accompany proprietary names, eg ©,  ®, or TM  (e.g. next to drug or reagent names). Therefore please remove all instances of trademark/copyright symbols throughout the text, including:

- ® on page: 4

- TM on page: 40.

5) Thank you for including an Ethics Statement for your study. Please include:

i) The full name(s) of the Institutional Review Board(s) or Ethics Committee(s).

6) Please upload all main figures as separate Figure files in .tif or .eps format. For more information about how to convert and format your figure files please see our guidelines: 

7) We notice that your supplementary Table is included in the manuscript file. Please remove it and upload it with the file type 'Supporting Information'. Please ensure that each Supporting Information file has a legend listed in the manuscript after the references list.

8) Some material included in your submission may be copyrighted. According to PLOSu2019s copyright policy, authors who use figures or other material (e.g., graphics, clipart, maps) from another author or copyright holder must demonstrate or obtain permission to publish this material under the Creative Commons Attribution 4.0 International (CC BY 4.0) License used by PLOS journals. Please closely review the details of PLOSu2019s copyright requirements here: PLOS Licenses and Copyright. If you need to request permissions from a copyright holder, you may use PLOS's Copyright Content Permission form.

Potential Copyright Issues:

i) Figure 1a. Please confirm whether you drew the images / clip-art within the figure panels by hand. If you did not draw the images, please provide (a) a link to the source of the images or icons and their license / terms of use; or (b) written permission from the copyright holder to publish the images or icons under our CC BY 4.0 license. Alternatively, you may replace the images with open source alternatives. See these open source resources you may use to replace images / clip-art:

**Reviewers' comments:**

Reviewer's Responses to Questions

Reviewer #1: This manuscript presents a thorough analysis of the developmental dynamics of H2AK119 ubiquitination (K119ub) and H3K27 monomethylation (K27me1) in relation to both active and repressive chromatin marks. While the study is primarily descriptive, it offers novel insights into the potential functions of two underexplored chromatin modifications, particularly within a neurodevelopmental framework. The authors take advantage of the cellular homogeneity provided by the predominance of granule neurons in the developing (P12) and mature (3-month-old) cerebellum to investigate the genome-wide distribution of K119ub and K27me1. Using Cut&Run, they analyze these marks alongside enhancer-, promoter-, and heterochromatin-associated histone modifications. Notably, the authors creatively integrate a range of computational and statistical approaches to enhance the interpretation of these complex datasets. Their analyses reveal that the association of K119ub and K27me1 with active and repressive chromatin regions is both context- and developmental stage–dependent, and substantially more dynamic than previously suggested by studies focused on the canonical PRC1/2 complexes and their associated mark, H3K27me3. In particular, K119ub is enriched in regions that lose active promoter and enhancer marks during cerebellar maturation, whereas K27me1 is broadly distributed across euchromatic regions with no clear association with transcriptional activation or repression, nor predictive value for future transcriptional states.

The findings are significant, not only advancing our understanding of chromatin regulation in neural development, but also introducing computational and statistical methodological innovations that will likely be valuable for future studies—especially those involving dynamic, multifactorial chromatin landscapes.

While the integration of transcriptomic data could be more fully developed, I acknowledge the inherent challenges in aligning transcriptional outputs with chromatin dynamics, due to differences in sensitivity and variability. Therefore, I offer only minor suggestions aimed at improving the clarity of the manuscript and interpretation of certain figures.

Minor Comments:

1-The change on K119ub abundance from early to late cerebellum on each of the early and late eu- or heterochromatin marks is very similar (within each mark). Is this just a coincidence or due to similar peaks identified for each of the marks in early and late cerebellum (i.e. how different/similar are the H3K27me3 peaks between early and late cerebellum)?

2-What is the transcriptional status of the bivalent promoters with increased abundance of K119ub from early to late cerebellum? Does this differ from the promoters that only have H3K4me3?

3-While H3K27ac is associated with active enhancers, there are instances of H3K27ac in poised or inactive enhancers as well. How certain are the authors that the increase of K119ub from early to late cerebellum in clusters of H3K27ac enhancer regions (Fig 4E-F) is associated with transcriptionally active genes? These figures would also benefit from showing the IGV screenshot of H3K27ac in late cerebellum.

4-Although it does not seem to be a correlation between K27me1 and RNA abundance in late cerebellum. Can authors test the early and late expression levels of the genes that gain K27me1 across development vs the ones that lose K27me1? (top 10% and bettom 10% in Fig 9D).

5-It is unclear to me why the number of peaks for each mark is different in the genome-wide aggregate heatmaps and in the volcano and log2-log2 plots in Fig 2-4. For instance, why are there 36,142 H3K27me3 peak summits in early cerebellum in Fig 2B and 12452 H3K27me3 peak regions in early cerebellum in Fig 2C,D? Maybe worth adding a note in the figure legend: i.e. note that number of peaks differe between Fig2B and C,D due to the use of different tools for peak definition (MACS2 narrow vs SICER).

6-It is unclear what is the difference between Fig 4B and Fig S6A and Fig 4C and S6B. Do perhaps the aggregated heatmaps in Fig S6A, B correspond to the marks in late cerebellum (instead of early, as indicated)?

7-Bar graphs in Fig S7 seem accidentally displaced downwards.

8-Can authors show pie chart of fractions of H3K27me1 peaks localized in different genome regions? This would support the statement in lines 429-431

9-There are few typos need to be corrected:

-Line162, ‘…early cerebellar confirmed…’ should be ‘…. early cerebellum confimed…’.

-Line 183, one of the two H3K27ac should be H3K9ac.

-The panels described in Fig 3 legend start from (E), should start from (A).

10-Details for computational analysis in the methods, are sometimes limited. Depositing the code generated for many of the analysis in GitHub, and referencing it in the final version of the manuscript would be convenient for others, like myself.

Reviewer #2: Srinivasan and Parmar et al claim that H2AK119ub spreads to euchromatin, not just facultative heterochromatin, independently of PRC2 during the development of the cerebellum from the early to late stages. Indeed, as shown in Figure 2C,D, the developmental shift from early to late stages shows that H2AK119ub and H3K27me3 do not necessarily colocalize. This can be interpreted to mean that PRC1 and PRC2 begin to exert independent control. From this perspective, the question of which chromatin regions H2AK119ub extends its distribution to is highly interesting for exploring new functions of PRC1.

The most important observation in this study is shown in Figure 1E: at both Early and Late stages, there are two patterns of uH2A – one that colocalizes with H3K27me3 and another that colocalizes with H3K27ac. These two types of uH2A modifications are likely driven by completely separate mechanisms. Regarding the regions where uH2A colocalizes with H3K27ac, there are two points of novelty. First, it is highly probable that this represents a regulatory mechanism independent of the repressive mechanism that works in concert with PRC2, showing dynamics that cannot be explained solely by the traditional model of "H2AK119ub recruiting H3K27me3"(Rob, Mol Cell, 2019). The second point is that these regions are observed much more prominently in the Late stage of cerebellar development, suggesting that this PRC2-independent regulatory region propagates widely towards the later stages of development. The authors focus on this second point of novelty, demonstrating its localization distribution and functional significance using cerebellar development as a model. This is a important novel perspective in chromatin research.

However, because the authors analyze the peaks of uH2A in the early and late stages independently, it is difficult for me to imagine how uH2A and other histone modifications change coordinately or antagonistically over the developmental time course. This contributes to a very unclear structure. Furthermore, most of the data presented lacks transparency in its overall quantification. For example, while summarizing the trends and correlations of each histone modification in early/late stages with correlation coefficients, as shown in Figure 5, is very clear and novel in terms of data presentation and analysis methods, it is difficult to readily accept without showing visualized heatmaps or concrete track views for individual gene loci and regions. To allow reviewers to better understand how the data was compiled and led to specific conclusions, adding figures that visualize the data closer to its raw form is recommended. This would improve the study's transparency and enhance the reliability of the results.

From the above points, the authors' findings are highly novel and interesting for the field of Polycomb and chromatin research, but they are difficult to accept in their current state. Adding the following data and experiments would make the manuscript more acceptable.

Major Comments

•The authors conclude that uH2A distributes to euchromatin in the later stages, based on CUT&RUN data for H3K27me3 and euchromatin histone modification marks. Figure 2 examines the movement of uH2A based on H3K27me3 peak calls, Figure 3 based on H3K4me3, and Figure 4 based on H3K27ac. However, the clearest way to present this would be to examine the results based on uH2A peak calls. The authors should analyze how the regions that colocalize with H3K27me3 change from Early to Late after calling uH2A peaks, and then re-analyze which histone modification patterns the uH2A-unique peaks coincide with. If the results are consistent with Figures 2-4, then Figures 2-4 could be moved to supplementary data.

•Regarding the four patterns of track views shown in Figure 1E, the authors should create a heatmap to show their genome-wide generality and indicate the number of regions for each pattern. Furthermore, for these four patterns, the authors should investigate whether the H3K27me3/H3K27ac levels in the Late stage all change similarly, or if they show diverse patterns of change. The results should be checked to confirm they are not inconsistent with Figures 6C and 6G.

•Observing data like Figures 1F,G, regions that initially had H3K4me3 or H3K9/27ac deposition and showed RNA transcription in the early phase appear to have increased H2AK119ub/H3K27me3 and suppressed transcription in the late stage. This appears to be simply repression by PRC1/2 in the late stage, rather than H2AK119ub spreading into Euchromatin. It is unclear where the novelty lies in this case. Are these loci not representative of the authors' main claim? It is more important to show the CUT&RUN data for each histone modification shown in Figure 1E for both Early and Late stages.

•As shown in Figure 6G, uH2A increases in all clusters regardless of changes in other histone modifications. This could be interpreted as simply a result of clustering based on the uH2A peak calls in the Late stage. Indeed, Figure 6B shows regions where uH2A decreases. This issue arises because independent populations are analyzed, which can cause confusion for the reader. Therefore, the authors should show how the number and movement of uH2A peaks change from early to late using Venn diagrams or similar based on the peak call results for both Early and Late stages. Based on this, it would be clearer to divide regions into those where uH2A does not change from early to late, those where it decreases, and those where it increases, and present this as a heatmap. If, among these changes, there are regions where uH2A changes not only coordinately with H3K27me3 but also with euchromatin marks, or if the binding of uH2A in such diverse chromatin regions increases in later development, the authors' claim that uH2A propagates to euchromatin regions during cerebellar development would be convincing to everyone.

•Is the chromatin remodeling of uH2A and H3K27me1 entirely restricted to CGI regions? Is it possible to explain the PRC2-dependent and independent forms of uH2A by differences in the presence, density, or length of CGIs?

Minor Comments

•In Figure 1E, including CUT&RUN data for H3K27ac and H3K27me3 in the Late stage as well, and showing how the chromatin state changes in these regions between Early and Late stages before showing the uH2A changes would be clearer.

•In figures like Figure 8C and Figure 9B, the correlation coefficient results should be shown in the figures.

•Combining Figures 8A and 8B to show track views of K27me1, K27me3, K4me3, and RNA for both Early and Late stages would be clearer. Also, at this time, is uH2A increasing?

•In Figure 8 and beyond, where early H3K27me1 is broadly and uniformly distributed within and outside genes, and later becomes significantly enriched within specific gene bodies, what is happening with uH2A? If both H3K27me1 and uH2A are increasing, could it be interpreted that one reason for the decrease in H3K27me3 colocalization with uH2A is that uH2A is instead associated with H3K27me1?

•Are the data regarding the distribution of uH2A in the first half of the paper and the data on H3K27me1 being concentrated in specific genes discussed as separate mechanisms?

Reviewer #3: This is an interesting manuscript reporting compelling chromatin data that reveal new ways of thinking about polycomb complex functions in neuronal differentiation. The authors use the relative homogeneity of cell types in the mouse cerebellum to study how changes in distinct histone modifications representing the function of different PRC components change over the course of cellular differentiation. They are especially interested in the H2AK119ub mark deposited by the PRC1 complex, which has traditionally been associated with gene repression. Here as they characterize the distribution of this modification across chromatin in immature (P12) versus mature (3 months) cerebellum compared with other well characterized histone marks of chromatin compartments. They find that the distribution of H2AK119ub significantly colocalizes with the PRC2 induced mark H3K27me3 in P12 brain, consistent with a role in facultative heterochromatin. However they also find H2AK119ub at some active enhancers (H3K27ac+/H3K4me1+) suggesting that it might have non-canonical functions in euchromatin. By 3 months the distribution of the two marks diverges with H2AK119ub levels rising at regulatory elements in euchromatin and falling in facultative heterochromatin. These data enrich understanding of PRC1 functions that are independent of PRC2. The data are presented and evaluated in rigorous fashion – this was one of the best descriptions of bioinformatics choices for pipelines and normalization strategies that I have seen. I have only minor concerns.

1) The authors do quite a bit of filtering for peaks that have a histone mark with known meaning (such as H3K27me3 for facultative heterochromatin or H3K4me1 for poised enhancers) and then assess H2AK119ub with respect to these other chromatin domains. However it would be useful if they would also just show an overall MA plot of H2AK119ub changes between the two time points. Summary statistics for the genome distribution of this mark (exon/intron/etc) as well as the overall correlation with changing gene expression would be useful as well. These can kind of be derived from other data presented, but the manuscript would benefit from starting with data on this mark alone before proceeding to the comparisons.

2) Line 179 “there was no consistent pattern of enrichment or depletion…” That is not really true that there is no pattern, there is just more than 1 so perhaps saying no singular pattern rather than consistent?

3) There are multiple H2A variant histones – does the composition of histone variants change over time in these neurons and is there any difference in the regulation of different variants by ubiquitination at K119? If it is relevant, a comment could be added to the discussion.

4) Atoh1 is an odd choice to show as a bivalent gene at P12. This gene is expressed much earlier in cerebellar development and has largely turned off by P12, whereas bivalent genes are usually discussed in the context of genes poised for activation. One possible concern is that P12 could actually be a mixture of developmental cell states. Depending on the strain, granule neuron progenitors (GNPs) can still be present in the second postnatal week, along with a mixture of newborn and maturing CGNs. In that case the apparent bivalency might instead reflect H3K4me3 at this promoter in GNPs and H3K27me3 at the promoter in CGNs rather than true bivalency in the same cells.

5) Given that many sites of H3K4me3 and H3K27me3 do not change between P12 and 3 months, it would be useful in some of the figures to subset dynamic versus static peaks to see how these compare against the changing H2AK119ub distributions.

6) The CpG islands are called “promoter-associated” but S5A clearly shows one that is intragenic. Did the authors use all CpG islands or only those in certain geneic regions?

7) Figure 4G is not mentioned in the results.

8) Line 303 the phrase “rather closely mimicked” is difficult to interpret. The figure shows overlap at some sites, but is there a way to quantify the similarity? This may be in Figure S7 – this was difficult to tell because the formatting of S7 is off in the downloaded version.

9) I am confused about line 311 where the authors talk about PRC1 being redistributed toward enhancers at the neural progenitor stage. This comes after the authors conclude that between P12 and 3 months that H2AK119ub is gained in euchromatin in CGNs. However if there are neural progenitors in cerebellum they are certainly there at P12 not 3 months. I think the authors are probably suggesting that in ref 43 the NPC stage is the more mature stage compared with something else, but this was unclear.

10) Line 362 the authors cannot really say PRC1 shifts its “regulation” – they show it shifts its “distribution” via the distribution of the mark.

11) Line 364 the authors cannot say that H2K119ub and H3K27me3 “operate” independently at “most” loci in the cerebellum. They did not show this. They show that places where H2K119ub increases are not places that H3K27me3increases. They do not prove a lack of overlap. And definitely they cannot say anything about function.

12) Line 364 I do not see data that support the statement that “H2AK119ub localizes mainly to enhancer regions”.

13) Dividing versus non-dividing cells have fundamental differences in the ways the regulate distribution of their chromatin marks. For the cerebellum vs liver vs kidney comparison, what percentage of the kidney and liver cells are mitotic? Did the authors consider comparing their P12 cerebellar data to P4 or P7 cerebellum (or isolated granule neuron progenitors) where there would be more mitotic cells but all in the CGN lineage?

14) Are the GO terms in Fig 7G-H for the closest gene or did the authors consider only promoter/intragenic peaks that were easily mapped to a gene?

15) Regarding the plasticity of H3K27me3 distributions in developing CGNs, and the role of EZH1 and EZH2, the authors would do well to contextualize their data with other studies. A prior study already reported the regulation of H3K27me3 at multiple stages of CGN differentiation (PMID: 37092728) and that study should be cited here. Both that study and Ref 71 in the reference list used pharmacological inhibition of EZH1 or EZH2 to ask about the roles of these two enzymes in H3K27me3 distributions in CGNs and found roles for both. Finally with respect to the question of deposition versus maintenance of H3K27me3, the authors may want to check out PMID: 27526204, which showed that knockout of both Ezh1 and Ezh2 in adult brain is required to lead to loss of H3K27me3, demonstrating a role for both enzymes in mature neurons.

16) Line 389 “(Figure 7A-5C)” Presumably this is a typo.

17) Figure S6J I am not sure that the shared fraction can be described as a “sizable minority” when for H2AK119ub it appears to be about 45% in most of the comparisons. It seems like the authors should also show “shared” peaks among the three tissue types as well as combinations of “unique” peaks between two pairs to match the “cell-type specific” statement.

18) Line 849 “To Analysis”? Presumably this is a typo.

**Have all data underlying the figures and results presented in the manuscript been provided?**

Reviewer #1: **No: ** More details about the computational analyses generated or making Computational code public would be ideal.

Reviewer #2: Yes

Reviewer #3: Yes

PLOS authors have the option to publish the peer review history of their article (what does this mean? ). If published, this will include your full peer review and any attached files.

**Do you want your identity to be public for this peer review?** For information about this choice, including consent withdrawal, please see our Privacy Policy .

Reviewer #1: **Yes: ** Naiara Akizu

Reviewer #2: **Yes: ** Mio Harachi

Reviewer #3: No

**Figure resubmission:**
---

## [Decision Letter · Decision Letter 1]

18 Aug 2025

Dear Dr Ferguson,

We are pleased to inform you that your manuscript entitled "Polycomb repressive complexes 1 and 2 independently and dynamically regulate euchromatin during cerebellar neurodevelopment" has been editorially accepted for publication in PLOS Genetics. Congratulations!

Yours sincerely,

Marisa S Bartolomei

Academic Editor

PLOS Genetics

Marnie Blewitt

Section Editor

PLOS Genetics

Aimée Dudley

Editor-in-Chief

PLOS Genetics

Anne Goriely

Editor-in-Chief

PLOS Genetics

Comments from the reviewers (if applicable):

Reviewer's Responses to Questions

**Comments to the Authors:**

Reviewer #1: Authors have adequately addressed my and other reviewers comments and substantially improved the manuscript.

Reviewer #3: The authors have done an outstanding job of responding to all the reviewers. This is a significant work that advances understanding of neuronal differentiation.

**Have all data underlying the figures and results presented in the manuscript been provided?**

Reviewer #1: Yes

Reviewer #3: Yes

PLOS authors have the option to publish the peer review history of their article (what does this mean? ). If published, this will include your full peer review and any attached files.

**Do you want your identity to be public for this peer review?** For information about this choice, including consent withdrawal, please see our Privacy Policy .

Reviewer #1: **Yes: ** Naiara Akizu

Reviewer #3: No

**Data Deposition**

http://datadryad.org/submit?journalID=pgenetics&manu=PGENETICS-D-25-00481R1

**Press Queries**

---

## [Editor Report · Acceptance letter]

PGENETICS-D-25-00481R1

Polycomb repressive complexes 1 and 2 independently and dynamically regulate euchromatin during cerebellar neurodevelopment

Dear Dr Ferguson,

We are pleased to inform you that your manuscript entitled " 

Polycomb repressive complexes 1 and 2 independently and dynamically regulate euchromatin during cerebellar neurodevelopment" has been formally accepted for publication in PLOS Genetics! Your manuscript is now with our production department and you will be notified of the publication date in due course.

With kind regards,

Katalin Szabo

PLOS Genetics

On behalf of:
